# Functional clustering of dendritic activity during decision-making

**Aaron Kerlin[†]\*, Boaz Mohar[†], Daniel Flickinger, Bryan J MacLennan, Matthew B Dean, Courtney Davis, Nelson Spruston, Karel Svoboda**

Janelia Research Campus, Howard Hughes Medical Institute, Ashburn, United States

**Abstract** The active properties of dendrites can support local nonlinear operations, but previous imaging and electrophysiological measurements have produced conflicting views regarding the prevalence and selectivity of local nonlinearities in vivo. We imaged calcium signals in pyramidal cell dendrites in the motor cortex of mice performing a tactile decision task. A custom microscope allowed us to image the soma and up to 300 μm of contiguous dendrite at 15 Hz, while resolving individual spines. New analysis methods were used to estimate the frequency and spatial scales of activity in dendritic branches and spines. The majority of dendritic calcium transients were coincident with global events. However, task-associated calcium signals in dendrites and spines were compartmentalized by dendritic branching and clustered within branches over approximately 10 μm. Diverse behavior-related signals were intermingled and distributed throughout the dendritic arbor, potentially supporting a large learning capacity in individual neurons.

**\*For correspondence:**
akerlin@umn.edu

[†]These authors contributed equally to this work

## Introduction

Neurons are bombarded by information from thousands of synaptic inputs, which are sculpted by the active properties of dendrites (*Stuart and Spruston, 2015*). The role of active dendrites in single-neuron computation remains unclear. Active membrane conductances may simply counteract location-dependent disparities and passive sublinearities across synapses, producing neurons that integrate input in a point-like, linear manner (*Bernander et al., 1994*; *Cash and Yuste, 1999*; *Spencer and Kandel, 1961*). Alternatively, active conductances may add nonlinear operations to the input-output transformation of neurons. This could involve a single nonlinear operation (*Phillips et al., 2015*; *Takahashi et al., 2016*; *Ujfalussy et al., 2018*) or many subunits performing independent nonlinear operations within dendritic branches (*Poirazi et al., 2003*; *Polsky et al., 2004*). These computations determine the capacity of individual neurons to store and process information (*Poirazi and Mel, 2001*). The relevant form of dendritic integration could be determined not only by dendritic structure and ion channel expression, but also by the spatiotemporal pattern of input to the dendrite under behaviorally relevant conditions (*Polsky et al., 2009*; *Ujfalussy et al., 2018*).

Active membrane conductances can support different types of regenerative events that impact dendritic integration of input. These event types vary in location of initiation, primary ionic conductance, dynamics, and extent of their spread within the dendrite (*London and Häusser, 2005*; *Major et al., 2013*; *Spruston, 2008*). All of these events are associated with calcium influx. Back-propagating action potentials (bAPs) can generate widespread calcium transients, dependent on firing patterns and synaptic input (*Jaffe et al., 1992*; *Magee and Johnston, 1997*; *Spruston et al., 1995*; *Waters et al., 2003*). Calcium spikes initiated in the apical tuft by the conjunction of bAPs and tuft depolarization (*Kim and Connors, 1993*; *Larkum et al., 1999a*) generate large and reliable multibranch calcium transients (*Helmchen et al., 1999*; *Xu et al., 2012*). In contrast, local dendritic spikes can be initiated in individual thin branches and can be driven by electrogenesis produced by

NMDA receptors (*Branco and Häusser, 2011*; *Larkum et al., 2009*; *Major et al., 2008*; *Schiller et al., 2000*; *Wei et al., 2001*) and other voltage-gated conductances (*Ariav et al., 2003*; *Golding and Spruston, 1998*; *Losonczy and Magee, 2006*; *Milojkovic et al., 2005*; *Nevian et al., 2007*). In isolation, a local dendritic spike generates calcium influx that is maximal near sites of coactive synaptic input within its dendrite of origin while calcium influx is minimal or absent at other dendrites (*Major et al., 2008*; *Milojkovic et al., 2007*; *Schiller et al., 2000*; *Wei et al., 2001*). Local dendritic spikes can be facilitated (*Brandalise et al., 2016*) or suppressed (*Remy et al., 2009*) by other regenerative events, potentially producing complex patterns of depolarization and calcium accumulation across the dendritic arbor.

The functional role of local dendritic spikes remains unclear. Some studies report a high prevalence of local dendritic spikes during sensory stimulation (*Palmer et al., 2014*; *Smith et al., 2013*), although the majority of local dendritic spikes were accompanied by somatic spikes (*Palmer et al., 2014*) and exhibited a sensory selectivity identical to the somatic spikes (*Smith et al., 2013*). Other studies failed to find local dendritic spikes during sensory stimulation (*Svoboda et al., 1997*; *Svoboda et al., 1999*) or spontaneous activity (*Hill et al., 2013*).

Studies of local dendritic spikes during behavior have relied predominantly on calcium imaging. One study suggested that different behaviors trigger calcium transients in nonoverlapping tuft branches (*Cichon and Gan, 2015*), supporting independent local dendritic operations. However, another study found that calcium transients in the dendrites of CA1 neurons were almost always coincident with transients in the soma (*Sheffield and Dombeck, 2015*), implying that local dendritic spikes are rare or have a functional selectivity similar to the somatic output. Thus, the prevalence, independence, and selectivity of local dendritic activity during behavior remains uncertain.

Clustering of coactive inputs over small length scales could facilitate the generation of local dendritic spikes (*Losonczy and Magee, 2006*; *Major et al., 2008*; *Polsky et al., 2009*; *Weber et al., 2016*). Multiple in vivo studies have probed the selectivity of dendritic spine calcium signals in pyramidal neurons of primary sensory cortex; some of these studies support clustering of functionally similar inputs (*Scholl et al., 2017*; *Wilson et al., 2016*), while others do not (*Chen et al., 2011*; *Jia et al., 2010*; *Varga et al., 2011*). This discrepancy could reflect the details of the sensory features and species investigated; alternatively, differences in the methods used to disambiguate the contribution of bAPs, post-synaptic nonlinearities, and pre-synaptic input to spine calcium signals could complicate measurements of synaptic selectivity based on calcium imaging.

Here, we developed novel calcium imaging and analysis methods to estimate the spatial structure of activity in spines and dendrites while mice performed a decision-making task. Previous calcium imaging studies in dendrites during behavior have imaged short stretches of dendrite (*Cichon and Gan, 2015*; *Sheffield and Dombeck, 2015*), making it difficult to disambiguate global, branch-specific, or spine-specific activity. To address these issues, we constructed a custom random-access, high-resolution microscope that allowed us to simultaneously record calcium signals throughout a large part of the dendritic tree, while still resolving signals at the level of individual spines. We leverage these simultaneous recordings to correct for brain motion and estimate the spatial scales of dendritic activity, avoiding potential biases inherent in other measurements.

We imaged pyramidal neurons in the anterior lateral motor (ALM) cortex as mice performed a tactile decision-making task with three well-defined behavioral epochs: sensory sampling, planning (delay), and response (*Guo et al., 2014a*). Anterior lateral motor (ALM) cortex is critical for decision making and planned directional licking in rodents (*Guo et al., 2014b*; *Li et al., 2016*). Neurons within ALM (*Chen et al., 2017*; *Guo et al., 2014b*) and connected regions (*Guo et al., 2017*) exhibit diverse behavioral selectivity during a tactile decision task. We mapped the prevalence, selectivity, and organization of task-associated signals across the dendritic tree of individual neurons. We found that nearby spines and dendritic segments had similar behavioral selectivity, and that the branching structure of the dendritic tree compartmentalized task-associated calcium signals.

## Results

### High-resolution and large-scale dendritic calcium imaging during tactile decision-making

We imaged calcium-dependent fluorescence changes ('calcium transients') in the dendrites of GCaMP6f-expressing neurons in the anterior lateral motor (ALM) cortex of mice performing a tactile delayed-response task (*Guo et al., 2014b*). To characterize the spatial organization of task-related signals within dendritic arbors, it was critical to image large parts of the dendrite with high spatial and temporal resolution. We therefore constructed a two-photon laser-scanning microscope (*Denk and Svoboda, 1997*) that allows rapid (~15 Hz) imaging of the soma and contiguous dendrite in three dimensions, while resolving calcium transients in individual dendritic spines (*Figure 1A*). Two mirror galvanometers and a remote focusing system (*Botcherby et al., 2008*) steer 16 kHz scan lines (24 µm long) arbitrarily in three dimensions. The use of a voice-coil to move the remote mirror is similar to a large field of view two-photon mesoscope system (*Sofroniew et al., 2016*; numerical aperture = 0.6) however, the system described here provides high numerical aperture (1.05), defocuses the beam after the scanning elements, and provides roughly twice the lateral and axial resolution. The system provided a 0.35 µm lateral and 1.9 µm axial resolution in the center and 0.56 µm lateral and 4.0 µm axial resolution at the edges of a 525 µm x 525 µm x 300 µm imaging volume.

Stable and sparse neuronal labeling is required for high signal-to-noise ratio and accurate reconstructions of dendritic morphology. We used Cre driver lines with sparse expression in L2/3 of ALM (Syt17_NO14-Cre) or L5 (Chrna2_OE25-Cre; *Gerfen et al., 2013*). Chrna2_OE25-Cre mice expressed in a subpopulation of pyramidal tract (PT) neurons and not intratelencephalic (IT) neurons (*Figure 1—figure supplement 1*; *Gerfen et al., 2013*). These lines were crossed with a GCaMP6f reporter line (Ai93; *Madisen et al., 2015*) and tTa-expressing lines (see Materials and methods). Expression was sufficiently sparse and bright to allow reconstruction of dendritic arbors from two-photon anatomical stacks (*Figure 1B,F*). We first traced the dendritic arbors of individual neurons (*Figure 1B,F*). The morphological data were then imported by custom software for selecting dendritic branches for fast imaging. The software calculated imaging sequences that optimize the actuator trajectories to maximize speed and coverage (*Figure 1C,G*). We used iterative, non-rigid registration to correct recordings for motion in three dimensions (*Figure 1D,E,H,I*; *Figure 1—figure supplement 2*, *Figure 1—video 1*, and Materials and methods). These methods allowed us to record calcium transients in the soma, dendrites (up to 300 µm total length), and up to 150 spines.

Mice performed a whisker-based object localization task (*Guo et al., 2014b*). A pole was presented at one of two locations (for 1.25 s) and withdrawn; after a delay epoch (2 s), mice licked either a right or left lickport based on the previous pole location (*Figure 2A*). In L2/3 neurons, we imaged the soma or proximal apical dendrite as a reference for multi-branch ('global') events, likely mediated largely by bAPs (*Figure 2B*, *Figure 2—video 1*). For each imaging session (maximum one session per day; median duration: 61 min) we targeted dendrites that were not previously imaged. We imaged a total of 3728 spines on 11.4 mm of dendrite from 14 L2/3 neurons (seven mice; 52 behavioral sessions; per-session medians: 241 trials, 74 spines, 221 µm of dendrite). In L5 neurons, we imaged the apical trunk as a reference for global events in the apical tuft (*Figure 2C*, *Figure 2—video 2*). We imaged a total of 655 spines on 3.9 mm of dendrite from five L5 neurons (four mice; 16 behavioral sessions; per-session medians: 276 trials, 39 spines, 259 µm of dendrite).

### The majority of dendritic calcium transients are coincident with global events

Our imaging approach provided a map of calcium transients across the dendritic arbor during behavior, while simultaneously imaged the soma, where signals reflect action potentials, with negligible contributions from subthreshold calcium signals (*Berger et al., 2007*; *Svoboda et al., 1997*; *Figure 2B*). In some instances, we observed activity restricted to single spines (*Figure 2B*, time-point i) as well as activity restricted to isolated dendritic branches, in the absence of detectable activity in the proximal apical dendrite of a L2/3 neuron (*Figure 2—figure supplement 1*), or the apical trunk of a L5 neuron (*Figure 2C*, time-point ii). However, these isolated dendritic events were rare, as most events were 'global', in that calcium transients were detected simultaneously throughout the

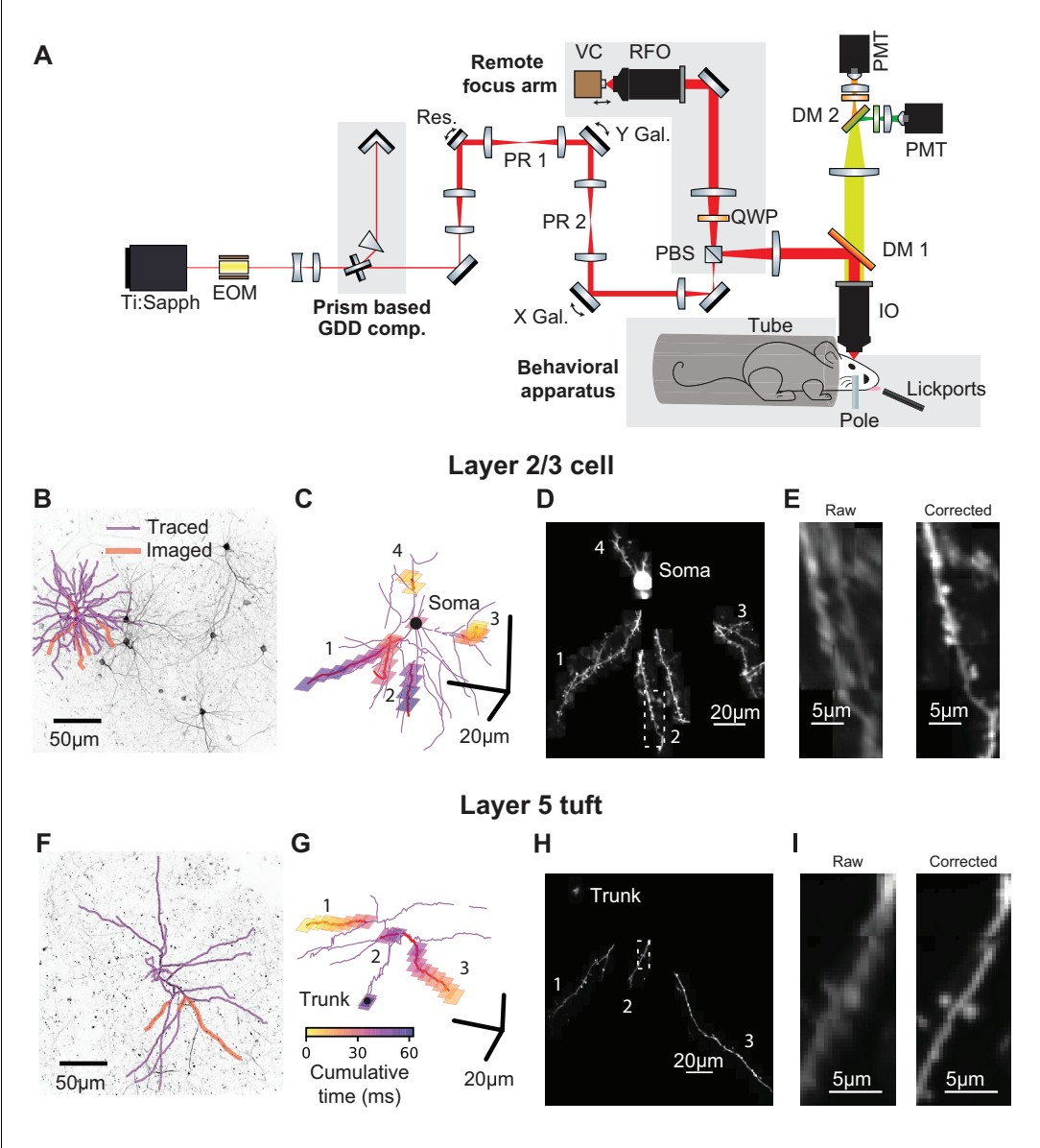

**Figure 1.** Targeted high-speed imaging in behaving mice. (**A**) Optical layout for high-speed, high-resolution imaging in three dimensions. An x-axis mirror galvanometer, remote focusing arm, and prism-based GDD compensation unit were added to a high resolution (NA = 1.0) resonant two photon microscope. EOM, electro-optic modulator; GDD, group delay dispersion; Res., 8 kHz resonant scanner; PR, pupil relay; Gal., galvanometer; PBS, polarizing beam splitter; QWP, quarter wave plate; RFO, remote focusing objective; VC, voice coil; DM, dichroic mirror; IO, imaging objective; PMT, photomultiplier tube. (**B**) Maximum intensity projections (MIP) of anatomical stack collected from Syt17-Cre x Ai93 (pia to 306 um depth) mice. Traced dendrite (purple lines) and example targets (red lines) for an example imaging session. (**C**) Spatial and temporal distribution of the frames that compose the example functional imaging sequences in (**B**). (**D**) Average MIP of 30 min of the functional imaging sequence shown in (**B, C**). (**E**) Close-up of the dendritic branch outlined in (**D**) before and after motion correction. (**F–I**) same as (**B–E**) for a layer 5 cell (MIP in (**F**) is pia to 560 um depth). See also *Figure 1—figure supplement 1* for characterization of the transgenic lines and *Figure 1—figure supplement 2* for details on motion registration. The online version of this article includes the following video and figure supplement(s) for figure 1:

**Figure supplement 1.** Histological characterization of transgenic mouse lines.

**Figure supplement 2.** Image regestration example.

**Figure 1—video 1.** Registration Example.

https://elifesciences.org/articles/46966#fig1video1

soma (or proximal dendrite) and all of the imaged parts of the dendritic arbor (*Figure 2B*, time-point

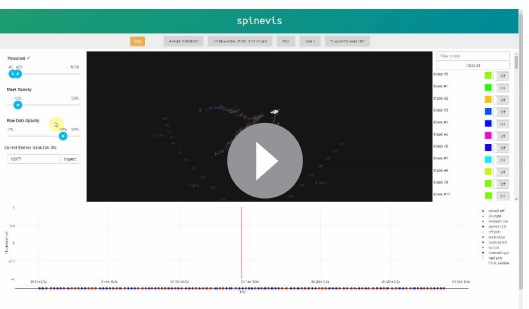

**Video 1.** Exploring the data online with SpineVis. In this screencast we show how to use the SpineVis website to look at the data in *Figure 2B*. On top center is the main viewing area where dragging will change the view in 3D. Clicking on a mask in that window will pull up the fluorescence trace for it in the lower window. The lower window has zoom and pan abilities that are linked to the upper window displaying the timepoint indicated by the black line in the center. To the left are display controls for changing lookup table values and opacity, followed by a timepoint selection window. To the right is the mask lookup window. Below the florescence trace are markers indicating behavioral events (e.g., blue triangle is a lick right event).

https://elifesciences.org/articles/46966#video1

ii; *Figure 2C*, time-point i).

Detecting local dendritic events could be limited by the signal-to-noise ratio of our measurements. Although ex vivo (*Golding et al., 2002*) and in vivo (*Svoboda et al., 1999*) experiments indicate that the calcium influx triggered by local regenerative dendritic events is often larger than the influx triggered by bAPs, we avoided assumptions about the magnitude or discrete nature of local events in dendrites during behavior. To estimate the prevalence of local events, we calculated a sample-by-sample probability that the global reference (soma or apical trunk) was below — and the dendrite above — a range of ΔF/F thresholds, while accounting for measurement noise (see Materials and methods, *Figure 3A*, *Figure 3—figure supplement 1*). We used these probabilities to estimate the proportion of activity that was independent of (i.e., not coincident with) the global reference. We also performed this analysis on spines. We limited our L2/3 data to sessions with simultaneous recording from the soma (234 dendritic segments, 30 μm long; 1625 spines). All of our L5 tuft recordings included simultaneous measurements from the apical trunk.

In L2/3 dendrites, independent dendritic activity was rare. For example, with the somatic threshold set to detecting 1–2 spikes (~15% ΔF/F; *Chen et al., 2013*) the probability of independent activity in L2/3 dendrites barely rose above the expected false-positive rate across a range of ΔF/F thresholds (*Figure 3C*, mean difference at threshold of ΔF/F > 1: 0.037 ± 0.009). In contrast, the proportion of independent activity in spines was higher by one order of magnitude (*Figure 3B,C*; see *Figure 3—figure supplement 1* for complete co-active, independent, and false positive rate grids). The proportion of independent activity in spines was close to 30%, even at thresholds where the false positive rate approached zero (mean difference at threshold of ΔF/F > 1: 0.324 ± 0.006).

A higher proportion of independent dendrite activity was observed in L5 tufts, compared to L2/3 dendrites (*Figure 3C*, mean difference at threshold of ΔF/F > 1: 0.15 ± 0.02, p<10⁻¹² Wilcoxon rank-sum test), although the majority of activity was still coincident with the global signal measured in the apical trunk. The distribution of independent activity across individual dendrite segments was skewed, especially in L2/3 (*Figure 3D*), where the top 10% most independent segments accounted for 76% of all independent activity (versus 35% for L5 dendrite segments; p<0.001 K-S test on distributions). Independent activity in dendrites often corresponded to sustained elevation in fluorescence that began with a global event but outlasted the global event by 100 s of milliseconds or even several seconds (*Figure 2—figure supplement 1*; *Figure 3—figure supplement 2*). We note that these low rates of independent activity do not preclude local modulation of the amplitude of dendritic signals during global events, for example by local dendritic spiking or subthreshold NMDA receptor cooperativity that is coincident with global events.

## Task-related calcium signals in the dendrite

Individual dendrites of sensory cortex neurons receive inputs with diverse feature selectivity (*Chen et al., 2013*; *Chen et al., 2011*; *Jia et al., 2010*; *Scholl et al., 2017*; *Varga et al., 2011*; *Wilson et al., 2016*). Mapping the diversity of functional input to individual neurons in frontal cortex is critical to understand decision-making, motor planning, and short-term memory (*Inagaki et al., 2019*; *Li et al., 2016*). Synaptic input is reflected in local changes in dendritic and spine calcium. To characterize local components of dendritic activity we estimated and removed the bAP-related

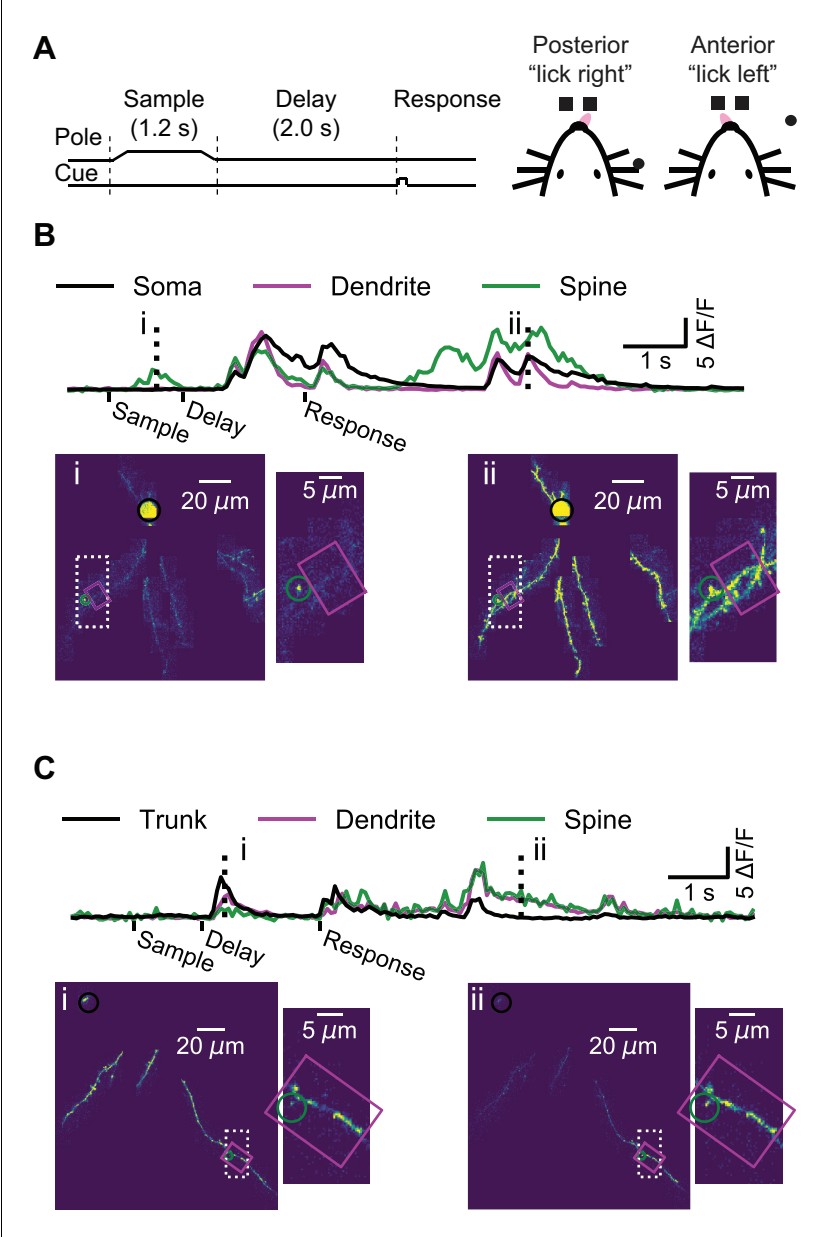

**Figure 2.** Dendrite and spine calcium activity. (**A**) Mice were trained to lick either a right or left target based on pole location. The pole was within reach of the whiskers during the sample epoch. Mice were trained to withhold licks until after a delay and auditory response cue. (**B**) Top, example somatic (black), dendritic (magenta), and spine (green) calcium signals from a layer 2/3 example session (as shown *Figure 1B–E*). Bottom, maximum intensity projections (au) and branch insets at selected times (dashed vertical lines in upper traces). Note independent spine activity at time i. (**C**) Same as (**B**) but for the layer five example session (as in *Figure 1F–I*). The apical trunk (black) was targeted as a reference for global activity. Note branch-specific sustained activity at time ii.

The online version of this article includes the following video and figure supplement(s) for figure 2:

**Figure supplement 1.** Examples of branch-specific persistent calcium activity.

**Figure 2—video 1.** Layer 2/3 Example Calcium Activity.

https://elifesciences.org/articles/46966#fig2video1

**Figure 2—video 2.** Layer 5 Example Calcium Activity.

https://elifesciences.org/articles/46966#fig2video2

component of calcium transients (*Figure 4—figure supplement 1*). Task-associated calcium transients in dendritic spines were consistent across behavioral trials (*Figure 4A–D*). Prior to removal of

the bAP component, trial-averaged spine activity was typically high during epochs when soma activity was also high (*Figure 4A,B*, spine i; *Figure 4E*), but some spine responses were independent of soma activity (*Figure 4C,D*; spine ii). Removal of the bAP-related component sharpened (*Figure 4A, B*; spine ii; see *Figure 4—figure supplement 2* for dendrite segment example), eliminated (*Figure 4A,B*; spine i), or had no effect (*Figure 4C,D*; spine ii) on the apparent selectivity of individual spines for specific trial epochs. After subtraction, the distribution of trial-averaged spine activity was less restricted to epochs with somatic activity (*Figure 4F*), confirming the existence of spine activity independent of bAPs.

To obtain a one-dimensional measure of the selectivity of responses for specific behavioral epochs, we treated the mean responses during sample, delay and response epochs as the magnitudes of three vectors separated by 120° in a polar coordinate system (*Figure 5*). The angle of the vector average then determined the epoch selectivity of each dendritic segment or spine (*Figure 5B*). To quantify and visualize selectivity for trial type, dendritic segments and spines were then given one of three colors depending on whether signals were selective for right (blue), left (red), or switched selectivity across epochs (purple; *Figure 5J*). In L2/3 dendrites (see *Figure 5A–C* for an example session), 63% of spines and 74% of short dendrite segments (~3 μm, see Materials and methods) exhibited significant (p<0.01, nonparametric ANOVA and standard error for epoch angle of <30 degrees) activity selective for specific trial epochs. 20% of spines and 30% of short dendrite segments exhibited significant (p<0.05, permutation test with Bonferroni correction) selectivity for trial-type (right vs. left) during at least one of the epochs. Similar selectivity was observed in L5 tuft dendrites (see *Figure 5F–H* for an example session; for all sessions: epoch selective: 46% of spines and 39% of dendrite segments, trial-type selective: 27% of spines and 38% of dendrite segments). Similar proportions of spines and dendritic segments with selectivity were observed after bAP subtraction (L2/3, *Figure 5D,E*, epoch selective: 53% of spines, 67% of dendrite segments, trial-type selective: 18% of spines and 27% of dendrite segments; L5, *Figure 5I,J*, epoch selective: 47% of spines, 54% of dendrite segments, trial-type selective: 28% of spines and 35% of dendrite segments). Thus, both spines and dendritic segments of neurons in ALM exhibit task-related selectivity distinct from the selectivity of global responses.

To quantify the diversity of task-related calcium signals in the dendrite, we analyzed differences in selectivity between pairs of selective spines from the same neuron. The distributions of pairwise differences in epoch selectivity was left-skewed for spine pairs of both L2/3 (*Figure 5K*, mean: 30 deg., 95% CI: 23–37 deg., IQR: 28 deg.) and L5 neurons (*Figure 5L*, 57 deg., 95% CI: 45–71 deg, IQR: 106 deg.). Epoch selectivity was different (p<0.05, bootstrap across trials) for 27% of spine pairs in layer 2/3 dendrite and 43% of spine pairs in L5 tuft dendrite. bAP subtraction shifted measures of diversity in epoch selectivity slightly higher (*Figure 5N,O*,L2/3: mean: 43 deg., 95% CI: 34–52 deg., IQR: 47 deg., significantly different: 31%; L5 tuft: mean: 60 deg., 95% CI: 46–78 deg., IQR: 127 deg., significantly different: 44%). Spine pairs exhibited different trial-type selectivity at proportions similar to epoch selectivity for both L2/3 and L5 (*Figure 5M,P*). Similar diversity of epoch and trial-type selectivity was observed between pairs of dendrite segment (*Figure 5—figure supplement 1*). This analysis shows that within the dendritic tree of individual neurons in frontal cortex, intermingled spines and dendrite segments can have distinct selectivity.

## Limitations in unmixing local activity and back-propagating action potentials

Previous studies of the functional responses of dendritic spines estimated and removed the bAP component, using linear regression between signals in spines versus nearby dendrites (*Chen et al., 2013*; *Iacaruso et al., 2017*; *Wilson et al., 2016*). Computer simulations show that this approach produces biased estimates of correlations between nearby spines (*Figure 6—figure supplement 2*). This is because the depolarization required for calcium influx in one compartment also acts on nearby compartments. In addition, the subtraction procedure does not account for differences in the decay times of bAP-generated transients in the soma compared to the dendrites. To overcome these issues, we deconvolved the reference signal (soma or apical trunk; *Pnevmatikakis et al., 2016*; *Vogelstein et al., 2010*), determined the amplitude and exponential decay that best fit each dendritic segment or spine signal (when convolved with the reference signal), and subtracted this fit (*Figure 4—figure supplement 1*).

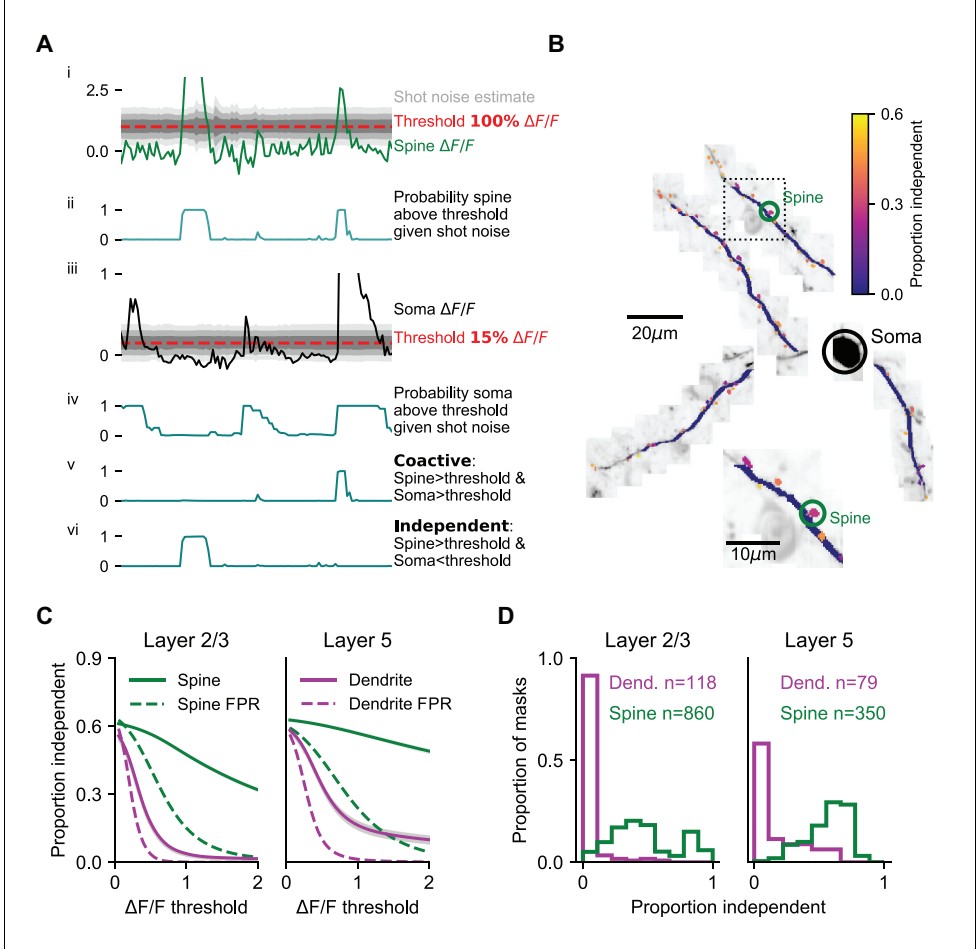

**Figure 3.** Coincidence of dendritic calcium transients with events in the soma and apical trunk. (**A**) Estimation of isolated spine activity as a function of threshold. (i) Example spine ΔF/F (green), threshold (red - 100% ΔF/F) and estimated uncertainty in ΔF/F due to shot-noise (gray shading). (ii) Probabilities of the spine to be above threshold. (iii, iv) same as (i, ii), but for the soma (black) with a lower threshold (15% ΔF/F). (v) Probability of co-activity. (vi) Probability of independent activity. (**B**) The proportion of independent activity in spines and dendrites with soma used as reference, example session. (**C**) Proportion independent as a function of threshold averaged across all spines (green) and dendrites (magenta). Left, L2/3 basal and apical dendrites with soma used as reference. Right, L5 tuft dendrites with apical trunk used as reference. Shaded region: SEM. Dotted lines: estimated false positive rate (FPR). (**D**) Distribution of the mean proportion independent activity of spines (green) and dendrites (magenta) of layer 2/3 cells (left) and L5 tuft (right). Note, L5 dendrites have a more rightward skewed distribution. See Figure S3 for the full distributions of co-active, independent and false positive rate as a function of thresholds. Layer 2/3: N = 6 mice; 13 neurons. Layer 5: N = 4 mice; five neurons.

The online version of this article includes the following figure supplement(s) for figure 3:

**Figure supplement 1.** Coincidence activity analysis with different thresholds.

**Figure supplement 2.** Excluding independent activity before or after reference activity.

Systematic errors in bAP-subtraction could affect the apparent organization of task-related calcium signals in the dendrite. Under- or over- subtraction sometimes produced inaccurate correlations between the bAP reference signal and spines as well as hypo- or hyper-diversity in the selectivity of spines (*Figure 6B*). To analyze the accuracy of various measures of dendritic calcium signals with our subtraction approach, we performed computer simulations with different assumptions about the processes underlying the spike-to-fluorescence transformation (*Figure 6—figure supplement 1*). We found that our subtraction procedure, as well as other linear subtraction methods we tested, failed to produce accurate estimates of correlations between spines and the global reference signal. Thus, we avoided a quantitative comparison of input (spine signals) and output (reference signals). The diversity of selectivity measured without subtraction (*Figure 5K,L*) provides a lower bound on the diversity of task-related selectivity. Simulations also indicated that signals

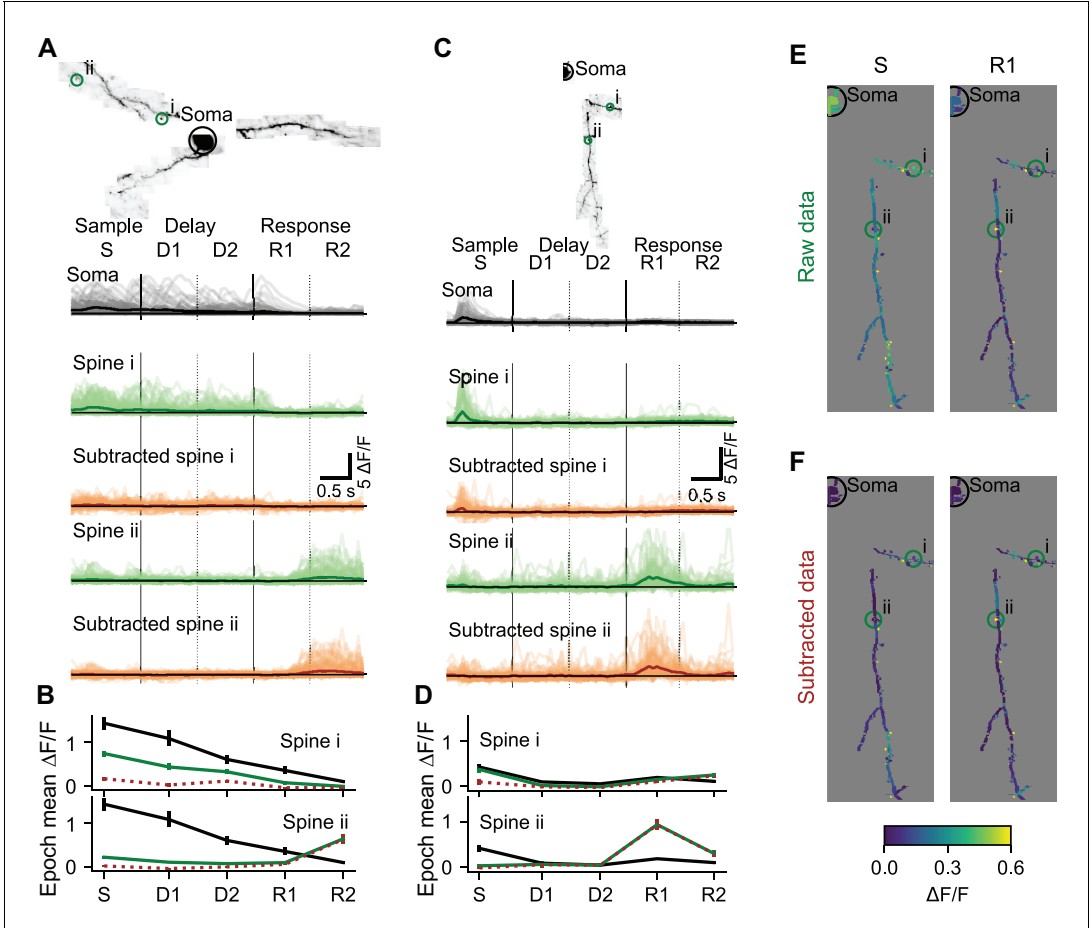

**Figure 4.** Local selectivity after removing the bAP component. (**A**) Subtraction of the bAP component from spine signals and estimation of trial average responses for two example spines. Image: MIP of an example L2/3 cell. Light lines: ΔF/F for all 110 correct right trials. Dark lines: trial-average ΔF/F. Black: Soma. Green: Spine before bAP subtraction. Brown: Spine after bAP subtraction. (**B**) Mean and standard error by trial epoch across all correct right trials of the spines in (**A**) with the same color-code. Note that most of the activity is being subtracted in spine i, but independent activity is not being subtracted in the response epoch of spine ii. (**C, D**) Same as (**A, B**) for a different L2/3 cell. (**E**) Trial-average responses of right sensory (**S**) and early response (**R1**) epochs for all dendrite segments and spines in the session shown in (**C**). (**F**) Same as (**E**) after removal of the estimated bAP component. See also *Figure 4—figure supplement 1* for an example subtraction of two spines.

The online version of this article includes the following figure supplement(s) for figure 4:

**Figure supplement 1.** Process for removing the estimated bAP-component of spine and dendritic segment signals.

**Figure supplement 2.** Example of independent and trial averaged activity in a L5 tuft dendrite segment.

processed with our approach to bAP-component removal produced accurate estimates of the spatial structure of dendritic activity (*Figure 6C*, *Figure 6—figure supplement 2*).

## Spatial clustering of task-related and trial-to-trial signals

Spatial clustering of coactive inputs could facilitate nonlinear computations within the dendrite (*Losonczy and Magee, 2006*; *Major et al., 2008*; *Polsky et al., 2009*; *Weber et al., 2016*). To quantify the similarity of selectivity as a function of distance along the dendrite, we calculated the correlation of average responses ('signal correlation') between pairs of spines and pairs of dendritic segments (*Figure 7A*). We randomly selected non-overlapping sets of trials to calculate trial-average responses of each dendritic segment or spine in the pair, which prevented trial-to-trial correlations ('noise correlations') from contaminating our estimates of signal correlation (*Cohen and Kohn, 2011*). Our imaging method allowed us to measure pairwise correlation from simultaneous recordings at distances considerably longer than previous studies (*Iacaruso et al., 2017*; *Wilson et al., 2016*). Pairwise correlations were strongest for nearby dendritic segments and spines in both L2/3

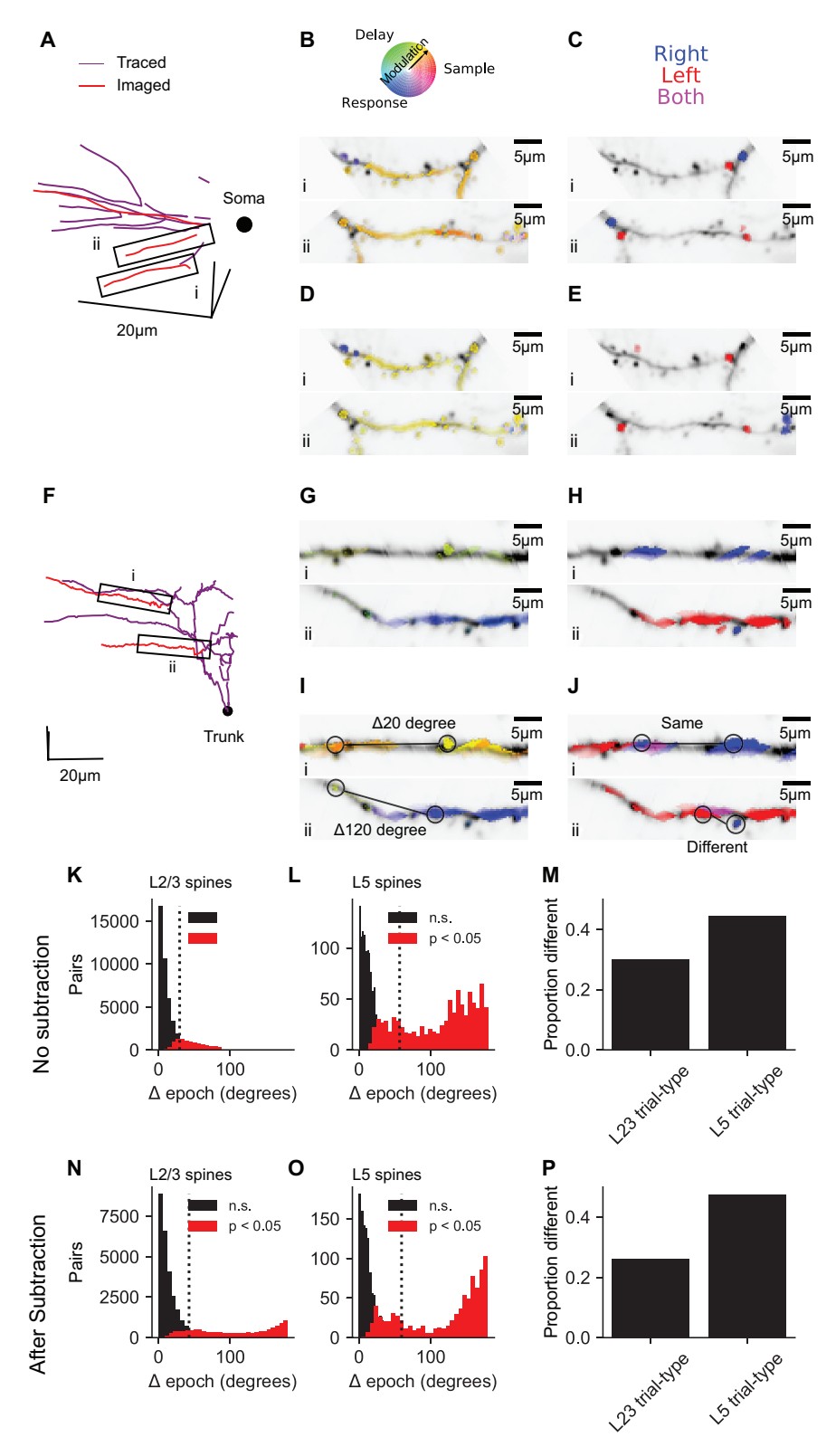

**Figure 5.** Dendrite and spine calcium signals exhibit diverse selectivity for trial epoch and trial type. (**A**) Location of simultaneously imaged dendrite (red lines) relative to the soma (black dot) and connecting dendrite (purple) that was not imaged for an example imaging session of a L2/3 cell. (**B**) Epoch selectivity for masks at two locations denoted by black boxes in (**A**). Mean sample, delay, and response epoch ΔF/F provided the magnitude for three vectors separated by 120°. The angle of the vector average in a polar RGB space determined the color of each mask. Only masks with significant

*Figure 5 continued on next page*

*Figure 5 continued*

epoch selectivity (nonparametric ANOVA, p<0.01 and epoch angle SE <30 degrees, see Materials and methods) are colored. (C) same as (B) but for trial-type selectivity. Masks significantly selective (permutation t-test, p<0.05 with Bonferroni correction) for right are blue, selective for left are red, and selective for both right and left depending on epoch are purple. (D, E) Same as (B, C), but with bAP subtraction. (F–J) Same as (A–E), but for an example L5 tuft session. Black dot in (F) denotes apical truck. (K) Distribution for L2/3 neurons of differences in epoch selectivity between epoch selective spines (epoch angle CI <30 degrees) on the same neuron for L2/3 neurons (N = 56568 pairs, 2038 spines). Red: Significantly different (p<0.05; bootstrap test on epoch angle). Black: Not significantly different. Dotted line: Mean angle across all pairs. (L) Same as (K), but for L5 tuft (N = 2060 pairs, 237 spines). (M) Of trial-type selective spines, proportion of spine pairs with different trial-type selectivity (L2/3: N = 16485 pairs, 905 spines; L5: N = 788 pairs, 135 spines). (N–P) Same as (K–M), but with bAP subtraction (N: N = 38677 pairs, 1692 spines; O: N = 1958 pairs, 235 spines; P,L2/3: 15284 pairs, 802 spines; P,L5: 7328 pairs, 143 spines).

The online version of this article includes the following figure supplement(s) for figure 5:

**Figure supplement 1.** Dendrite Signals Exhibit Diverse Selectivity for Trial Epoch and Trial Type.

---

dendrites and L5 tufts (*Figure 7B,C*; p<0.001 for both, nonparametric ANOVA comparison to shuffle). These signal correlations (especially in L2/3 dendrites) had a long linear decay in addition to a short exponential component (*Iacaruso et al., 2017*; *Wilson et al., 2016*). Fits to this exponential-linear function (see Materials and methods), provided estimates of the length constant of the exponential component that ranged from 7 to 19 μm (*Figure 7C*). Differences in exponential length constant between spine pairs, dendrite pairs, L2/3 neurons, and L5 neurons were not significant (p>0.05 for all comparisons, shuffles across sessions). Combining across cell types, we estimated a length constant of 8 ± 4 μm for dendrite segments and 13 ± 6 μm for spines.

In addition, we measured noise correlation among spines and dendrite segments. These noise correlations may reflect variable sensation and behavior during task performance, common sources of input, or other processes. As with signal correlations, we observed a strong effect of distance on pairwise noise correlations for dendrite segment pairs and spine pairs in L2/3 dendrites and L5 tufts (*Figure 7E,F*; p<0.001 for both, nonparametric ANOVA comparison to shuffle). Fits to the exponential-linear function provided estimates of the length constant of the exponential component that ranged from 9 to 18 μm (*Figure 7F*). Differences in exponential length constant between spine pairs, dendrite pairs, L2/3 neurons, and L5 neurons were not significant (p>0.05 for all comparisons, shuffles across sessions). Combining across cell types, we estimated a length constant of 10 ± 3 μm for dendrite segments and 14 ± 3 μm for spines. Thus, the spatial profile of noise correlations within the dendrite was not significantly different from the profile for signal correlations.

To control for artefactual correlations due to residual image motion, we analyzed correlations in pairs with short Euclidean distance (<15 μm) but long distance along the dendrite (>30 μm). In dendrites of L2/3 neurons – where a sufficient number of pairs met this criteria – pairs with short Euclidean distance had significantly lower correlation than pairs with an equivalent distance along the dendrite (*Figure 7C,F*, purple points), indicating that residual motion makes little contribution to the distance-dependent correlations.

## Dendritic branching compartmentalizes task-related calcium signals

Impedance mismatch at branch points (*Marlin and Carter, 2014*; *Müllner et al., 2015*) and branch-specific regulation of excitability (*Losonczy et al., 2008*) may compartmentalize signals to dendritic branches. To test if this influences behavior-related calcium signals, we compared the similarity of selectivity within and across branches. The distribution of epoch selectivity was clearly different from branch to branch in some imaging sessions (*Figure 8A*). We measured the mean signal correlation for spine and dendrite pairs within versus across branches. Branch location had a significant effect on spine and dendrite pairs from both L2/3 and L5 tuft (*Figure 8B*). This could reflect an influence of branch structure, the distance-dependence of signal correlations, or both. To selectively test for an influence of branch points, we restricted the data to pairs less than 10 μm apart that were either within a branch or crossed a single branch point. Correlations were lower when a branch point was crossed (*Figure 8C*; p<0.05 for all groups of pairs except L5 tuft spines).

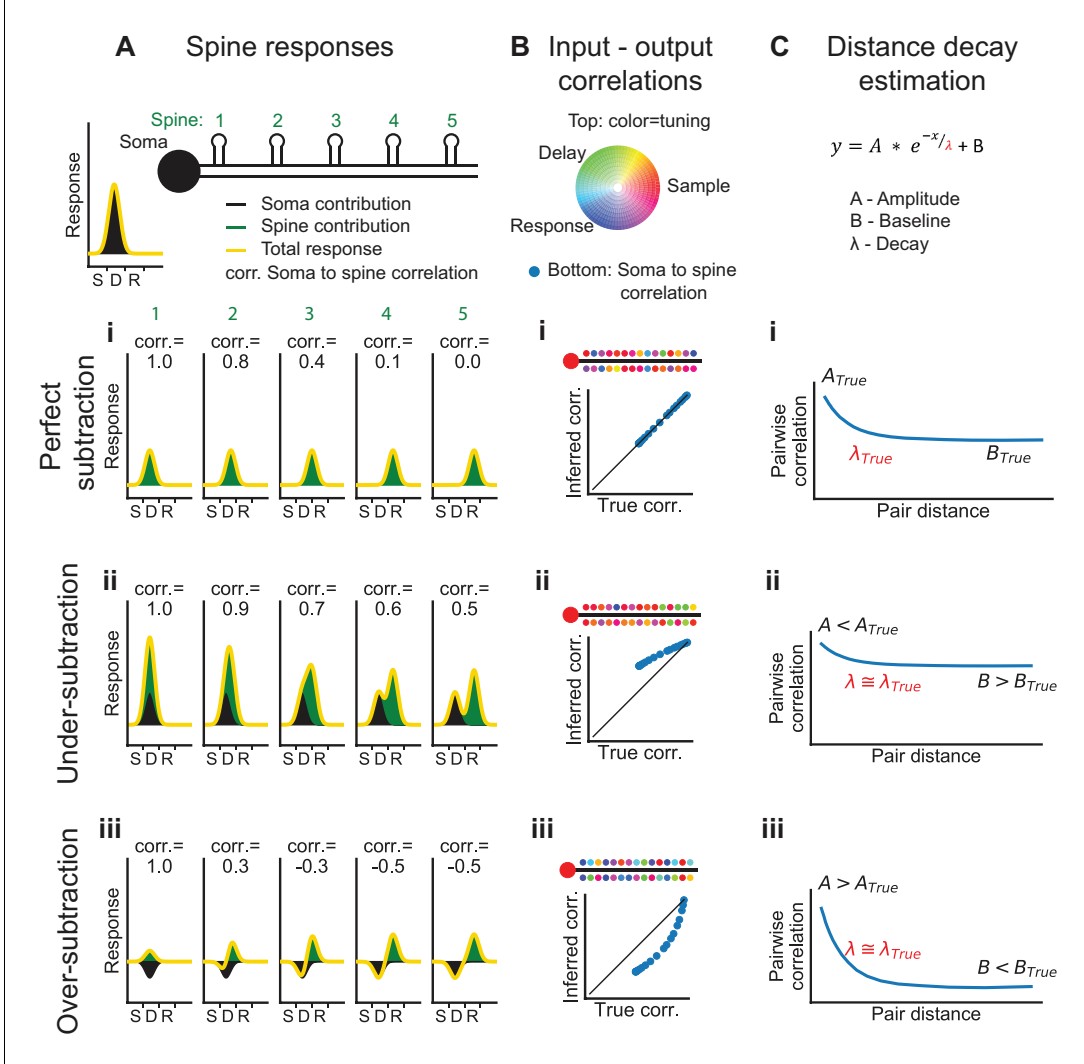

**Figure 6.** Interpretation of dendritic calcium signals before and after removal of the estimated contribution from back-propagating action potentials (bAPs). (**A**) Spine responses under different bAP subtraction regimes. Top, Cartoon of the soma and spatial organization of five spines. Soma trial-average response (black curve) is centered between the sample and delay epochs. (i) True (perfect bAP-component removal) tuning curves for the spines exhibiting a distance-dependent similarity of selectivity. (ii) If the bAP-component is under subtracted, subtracted tuning curves will still be biased towards the somatic selectivity. (iii) If the bAP-component is over subtracted, subtracted tuning curves will be biased away from the somatic selectivity. (**B**) Input-output correlation under different bAP subtraction regimes. Top, polar RGB representation of spine selectivity as in *Figure 5B*. (i) With perfect subtraction, the inferred correlation of each spine tuning curve with the somatic tuning curve matches the true correlation (plot). In this cartoon, spine selectivity is biased towards the selectivity of the soma (redder), but there is still significant diversity (green and blue spines). (ii) Under-subtraction of the bAP-component leads to less diverse spine selectivity and higher correlations with the somatic output. (iii) Over-subtraction of the bAP-component leads to more diverse spine selectivity and less correlation with the somatic output than truth. (**C**) Distance-dependent correlation between pairs of spines can be fit with a three-parameter exponential function. A: magnitude of distance-dependent correlations, λ: length constant, B: magnitude of distance-independent correlations. (i-iii) Different levels of subtraction dramatically shift the inferred values of A and B, but λ is accurately estimated. See also *Figure 6—figure supplement 1* for performance of input-output simulation and *Figure 6—figure supplement 2* for length constant simulations.

The online version of this article includes the following figure supplement(s) for figure 6:

**Figure supplement 1.** Simulating the effects of removing the estimated bAP-component on the inferred correlation between spines and the reference signal.

**Figure supplement 2.** Simulating the effects of removing the estimated bAP-component on the inferred length constant of correlation between spines.

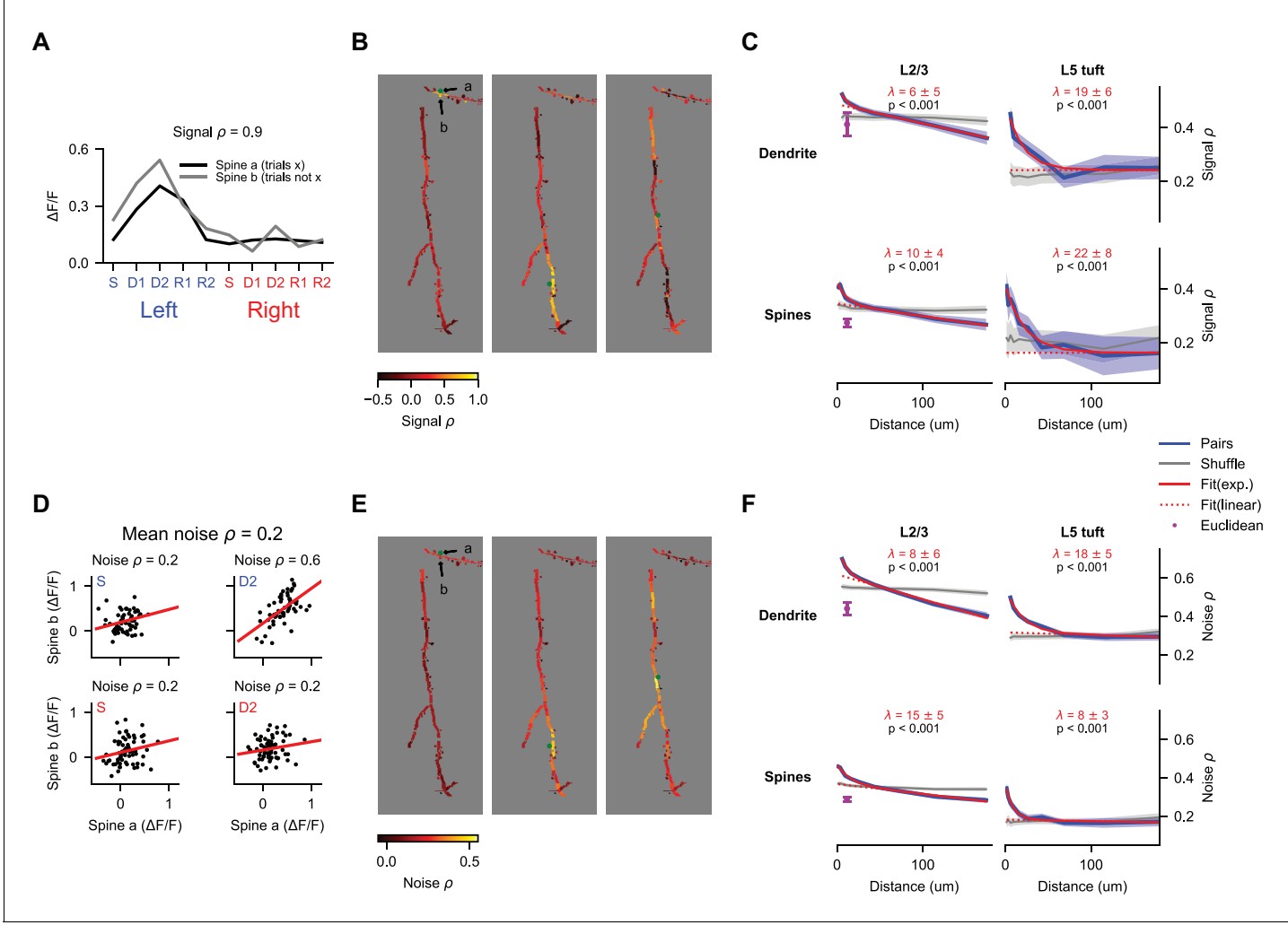

**Figure 7.** Behavior-related calcium signals are organized in a distance-dependent manner within the dendritic tree. (**A**) Estimation of pairwise signal correlation after bAP subtraction for two example spines (denoted in (**B**)). Trial average responses for each epoch and trial type are calculated for each mask from non-overlapping trials (to exclude noise correlations). Signal correlation is the Pearson correlation coefficient between these sets of responses. (**B**) Pairwise signal correlation of three spines (green dots) with all other masks in an example session (same session as shown in *Figure 4C*). (**C**) Pairwise signal correlation as a function of traversal distance through the dendrite (see Materials and methods for binning; L2/3 dendrite: N = 22688 pairs, 1533 segments; L5 dendrite: N = 7468 pairs, 432 segments; L2/3 spines: N = 45993 pairs, 2058 spines; L5 spines: N = 2783 pairs, 285 spines). Shaded regions are ± SEM (see Materials and methods). Magenta point is the mean pairwise correlation of masks with Euclidean distance < 15 μm and traversal distance > 30 μm. Only masks with significant (p < 0.01) task-associated selectivity were included. p-values from nonparametric comparison to shuffle. λ is the mean length constant ± SEM (**D**) Estimation of pairwise noise correlation for two example spines (denoted in (**B**)). Each panel is an example epoch denoted by the colored text in the upper left. Each black point is a trial. Noise correlation for a pair is the mean of the correlations calculated across all epochs. (**E**) Same as (**B**), but for noise correlation. (**F**) Same as (**C**), but for noise correlation. All N same as in (**C**).

## Discussion

We recorded activity in the dendrites and spines of motor cortex pyramidal neurons as mice performed a tactile discrimination task. The majority of calcium transients in the dendrites were coincident with global events. Events that were not coincident with global events were rare and occurred with higher frequency in the dendritic tufts of L5 neurons than in the dendrites of L2/3 neurons. However, the amplitudes of local calcium signals were modulated by task-related variables and spatially clustered within individual dendritic branches. Our data suggest that sensorimotor signals are compartmentalized within the dendrites of neurons in motor cortex, consistent with models in which branch-specific information enhances the computational or learning capacity of neural circuits (*Guerguiev et al., 2017*; *Poirazi and Mel, 2001*; *Urbanczik and Senn, 2014*; *Wu and Mel, 2009*).

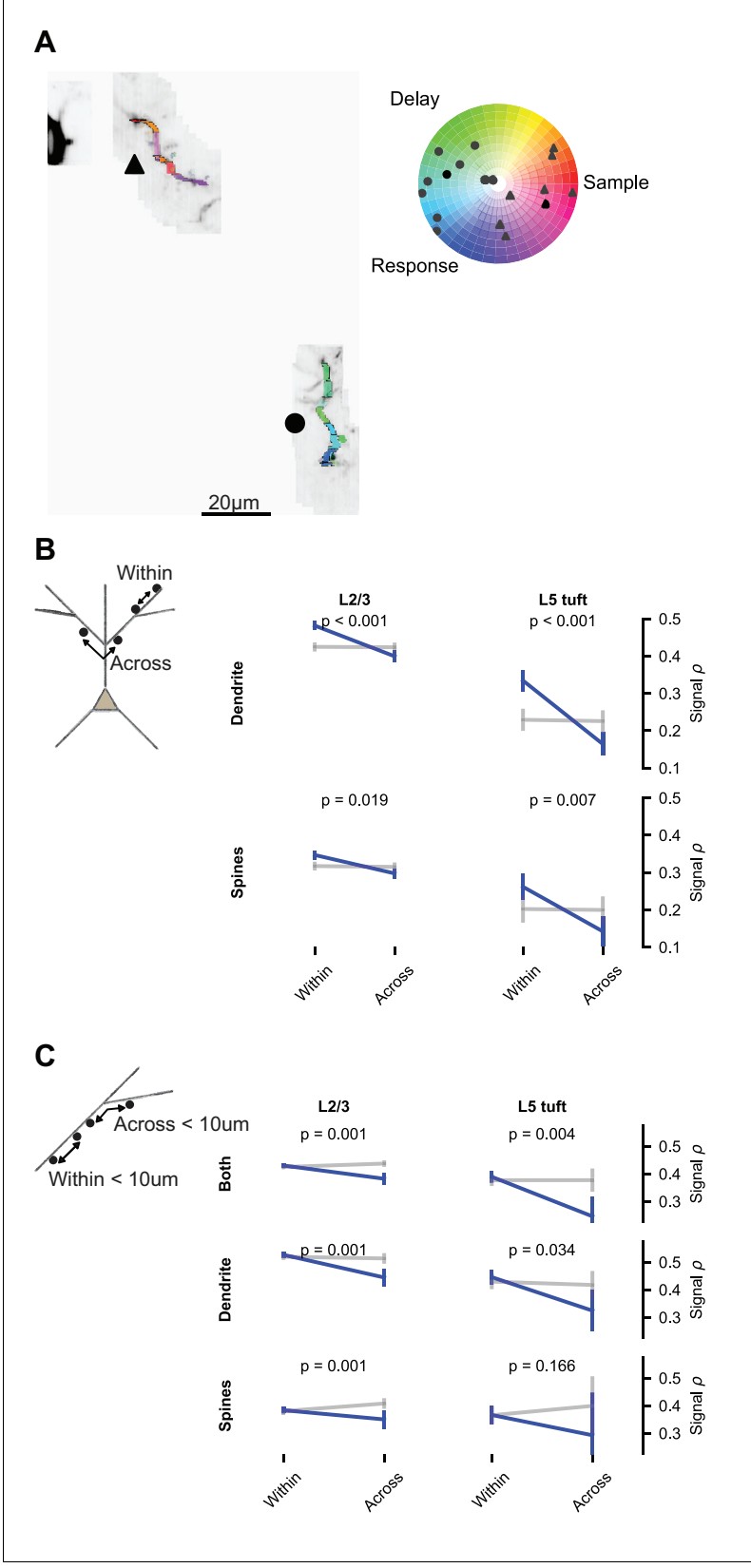

**Figure 8.** Dendritic branching compartmentalizes behavior-related calcium signals. (**A**) Example of clustering of epoch selectivity for a L2/3 session. Hue and saturation were determined for each mask as in *Figure 5B* after bAP subtraction. Markers (gray: individual masks, black: mean) in the polar plot denote the selectivity of all significantly

*Figure 8 continued on next page*

*Figure 8 continued*

selective (p<0.01) masks within the branches adjacent to the same marker in the colored MIP. (B) Pairwise signal correlations after bAP subtraction within versus across branches (blue: pairs; gray: shuffle), regardless of distance (L2/3 dendrite within: N = 8794 pairs, 1436 segments; L2/3 dendrite across: N = 13894 pairs, 1403 segments; L5 dendrite within: N = 2652 pairs, 406 segments; L5 dendrite across: N = 4816 pairs, 432 segments; L2/3 spines within: N = 16711 pairs, 2002 spines; L2/3 spines across: N = 29282 pairs, 1907 spines; L5 spines within: N = 1207 pairs, 269 spines; L5 spines across: N = 1576 pairs, 278 spines). (C) Short distance (<10 um) pairwise signal correlations within versus across a branch point (L2/3 both within: N = 10484 pairs, 3510 spines or segments; L2/3 both across: N = 1829 pairs, 814 spines or segments; L5 both within: N = 1368 pairs, 681 spines or segments; L5 both across: N = 139 pairs, 98 spines or segments; L2/3 dendrite within: N = 1533 pairs, 1392 segments; L2/3 dendrite across: N = 359 pairs, 323 segments; L5 dendrite within: N = 451 pairs, 376 segments; L5 dendrite across: N = 61 pairs, 62 segments; L2/3 spines within: N = 3834 pairs, 1929 spines; L2/3 spines across: N = 598 pairs, 400 spines; L5 spines within: N = 228 pairs, 233 spines; L5 spines across: N = 18 pairs, 20 spines). 'Both' includes dendrite-dendrite, spine-spine, and spine-dendrite pairs. All plots are mean and SE, p-values from nonparametric permutation test (each spine or dendrite segment drawn only once) comparison to shuffle.

## Simultaneous imaging of soma, contiguous dendrites, and spines

Our microscope and imaging strategy allowed us to obtain near-simultaneous, high-resolution 3D images of the soma and up to 300 µm of contiguous dendrite and spines at ~15 Hz. This provided several advantages over functional imaging of dendrites that intersect one or two imaging planes (*Cichon and Gan, 2015*; *Sheffield and Dombeck, 2015*). First, we were able to sample dendrites more efficiently. Second, imaging long stretches of contiguous dendrite allowed us to localize activity to specific dendritic regions (*Figure 2—figure supplement 1*; *Figures 7* and *8*). Third, averaging fluorescence across extended (30 µm) dendritic segments improved the SNR for our branch measurements (*Figure 3*) and reduced contamination of dendritic shaft signals by spine signals. Raw movies and processed traces for all figures can be browsed online (*Table 1*; *Video 1*).

For rapid scanning in 3D we used a resonant mirror, mirror galvanometer, and remote focusing (*Botcherby et al., 2008*). Actuator control signals were optimized computationally (see Materials and methods). This approach offers some advantages over acousto-optical scanning methods, which can have higher rates of dendrite and spine imaging (*Nadella et al., 2016*; *Szalay et al., 2016*), but a smaller field-of-view (FOV) and 1.5–3 times lower resolution depending on location in the FOV. Dense and high-resolution images of the targeted dendrite and nearby space were critical for identifying and excluding signals from crossing axons and boutons that would otherwise appear as strongly independent dendrite or spine activity (see Materials and methods). In the future, point-scanning systems combining motionless and mechanical deflection may provide the optimal trade-offs between speed, resolution, and FOV for dendrite and spine imaging (*Heberle et al., 2016*). Even higher imaging speeds can be obtained using alternatives to point-scanning, such as imaging with an elongated focus (*Lu et al., 2017*) or tomographic scans (*Kazemipour et al., 2019*), but at the expense of requiring higher average laser power and computational reconstruction of signal sources.

Identifying task-related activity in individual dendritic spines required recordings across 100 or more behavioral trials. Brain movement during task performance, especially during licking (*Andermann, 2010*; *Komiyama et al., 2010*), made stable spine recordings challenging. We addressed this issue by developing an iterative clustering and registration algorithm to obtain submicron, nonrigid registration (see *Figure 1—figure supplement 1*; *Figure 1—video 1*; Materials and methods).

## Analysis and interpretation of calcium signals in dendrites and spines

Understanding local dendritic operations will require precise measurements of synaptic activity and various post-synaptic processes. Calcium influx into the dendritic shaft and spines is produced by calcium-permeable receptors and voltage-gated calcium channels. Both types of conductances are modulated by synaptic currents and postsynaptic electrogenesis, complicating the interpretation of dendritic calcium signals. Synaptic signals can be isolated during dendritic spine imaging by abolishing bAPs, using invasive approaches that hyperpolarize (*Jia et al., 2010*; *Levy et al., 2012*) or depolarize (*Mainen et al., 1999*) the neuron. These manipulations, however, are difficult to apply in behaving animals, especially in chronic imaging preparations, and could trigger plasticity

**Table 1.** SpineVis online viewer links.

| Figure | Panel | Session name | Time points | Mask | Direct online viewer link |
|---|---|---|---|---|---|
| 1 | B-E | L23-session18 | | | http://spinevis.janelia.org/session/BMWR30:20151123:2:1:Layer23Session18 |
| 1 | F-I | L5-session3 | | | http://spinevis.janelia.org/session/BMWR58:20161220:1:2:Layer5Session3 |
| 2 | B | L23-session18 | Trace: 21520–21675<br>Image i: 21540<br>Image ii: 21625 | Spine: 27<br>Dendrite: 122<br>Soma: 119 | http://spinevis.janelia.org/session/BMWR30:20151123:2:1:Layer23Session18 |
| 2 | C | L5-session3 | Trace: 28040–28220<br>Image i: 28080<br>Image ii: 28158 | Spine: 36<br>Dendrite: 61<br>Trunk: 51 | http://spinevis.janelia.org/session/BMWR58:20161220:1:2:Layer5Session3 |
| 3 | A | L23-session20 | Trace: 21308–21367 | Spine: 48<br>Soma: 79 | http://spinevis.janelia.org/session/BMWR30:20151112:1:1:Layer23Session20 |
| 4 | A, B | L23-session21 | | Soma: 75<br>Spine i: 4<br>Spine II: 35 | http://spinevis.janelia.org/session/BMWR30:20151110:1:1:Layer23Session21 |
| 4 | C-F | L23-session13 | | Soma: 71<br>Spine i: 7<br>Spine II: 23 | http://spinevis.janelia.org/session/BMWR53:20161127:1:2:Layer23Session13 |
| 5 | A-E | L23-session5 | | | http://spinevis.janelia.org/session/BMWR47:20160711:1:1:Layer23Session5 |
| 5 | F-J | L5-session10 | | | http://spinevis.janelia.org/session/BMWR58:20170106:1:1:Layer5Session10 |
| 7 | A, B, D, E | L23-session13 | | Spine a: 2<br>Spine b: 1 | http://spinevis.janelia.org/session/BMWR53:20161127:1:2:Layer23Session13 |
| 8 | A | L23-session15 | | | http://spinevis.janelia.org/session/BMWR53:20161126:1:1:Layer23Session15 |
| F1-S2 | A-H | L23-session21 | | | http://spinevis.janelia.org/session/BMWR30:20151110:1:1:Layer23Session21 |
| F2-S1 | A | L23NoSoma-session20 | Trace: 43700–43900<br>Image i: 43793<br>Image ii: 43806 | Dendrite 1: 90<br>Dendrite 2: 85<br>Soma: 134 | http://spinevis.janelia.org/session/BMWR30:20151124:2:1:Layer23NoSomaSession20 |
| F2-S1 | B | L5-session3 | Trace: 6850–7050<br>Image i: 6937<br>Image ii: 6963 | Dendrite 1: 62<br>Dendrite 2: 57<br>Trunk: 51 | http://spinevis.janelia.org/session/BMWR58:20161220:1:2:Layer5Session3 |
| F2-S1 | C | L5-session12 | Trace: 46636–46836<br>Image i: 46724<br>Image ii: 46745 | Dendrite 1: 61<br>Dendrite 2: 67<br>Trunk: 59 | http://spinevis.janelia.org/session/BMWR59:20170104:1:1:Layer5Session12 |
| F2-S1 | D | L5-session12 | Trace: 49633–49833<br>Image i: 49723<br>Image ii: 49753 | Dendrite 1: 70<br>Dendrite 2: 66<br>Trunk: 62 | http://spinevis.janelia.org/session/BMWR59:20170104:1:1:Layer5Session12 |
| F4-S1 | A-G | L23-session18 | Trace: 20013–20075 | Spine 1: 20<br>Spine 2: 9<br>Soma: 119 | http://spinevis.janelia.org/session/BMWR30:20151123:2:1:Layer23Session18 |
| F4-S2 | A | L5-session10 | Trace: 34560–34900<br>Image i: 34616<br>Image ii: 34857 | Dendrite 1: 34<br>Dendrite 2: 38<br>Trunk: 36 | https://spinevis.janelia.org/session/BMWR58:20170106:1:1:Layer5Session10 |

(*Wigström et al., 1986*). Several studies (*Chen et al., 2013*; *Iacaruso et al., 2017*; *Scholl et al., 2017*; *Wilson et al., 2016*) have instead interpreted signals from nearby dendrites as a reference for computational removal of the bAP signal in spines. However, signals in the nearby dendrite are not necessarily exclusively or linearly related to bAPs: local synaptic input can generate coincident dendritic spikes (*Golding and Spruston, 1998*; *Losonczy and Magee, 2006*; *Schiller et al., 2000*), as well as amplify or suppress the amplitudes of bAP calcium transients in the dendrite (*Magee and Johnston, 1997*; *Remy et al., 2009*; *Waters and Helmchen, 2004*). To obtain a reference signal that more directly reflects action potentials, we used simultaneously recorded signals from the soma of L2/3 neurons, where subthreshold depolarization has a negligible impact on calcium signals (*Berger et al., 2007*; *Svoboda et al., 1997*). We developed a new bAP subtraction method that accounts for differences in calcium dynamics across compartments, especially the long decay dynamics in the soma compared to the dendrite.

Computer simulations indicated that conclusions drawn from this approach are limited (*Figure 6—figure supplement 1*). Inferred correlations between input (spines) and output (soma) did not reliably predict true correlations, even in simple and linear models of the relationship between activity and fluorescence signals in dendritic spines. This was also true for all linear subtraction methods we tested. Correlations at timescales shorter than the decay of the GCaMP signals were most affected. This is also the timescale over which synaptic input would be expected to influence somatic spiking, and therefore neuronal output. We therefore avoided drawing conclusions about how synaptic input might drive output.

Other measures of the spatial structure of dendritic activity, however, were accurately estimated using our bAP subtraction approach (*Figures 6* and *7* and *Figure 6—figure supplement 2*). We simulated pre-synaptic clustering or post-synaptic cooperativity with varying length scales, based on the real geometry and numbers of spines analyzed for L2/3 neurons (see *Figures 1B–E*, *2B*, *3B*, *4A*, *5A* and *8A* for example sessions). Using our subtraction methods on simulated data, the presence or absence of structured correlations was accurately detected and the inferred length scale of pairwise correlations closely matched the simulated scale. These estimates were robust to potential differences in the rate of calcium extrusion across different dendritic compartments, which was simulated with a distance-dependent variation in decay dynamics (see Materials and methods). The length scales we estimate could reflect pre- and post-synaptic processes and should thus be regarded as a description of the local component of calcium signals in the dendrite.

Incorporation of assumptions about the temporal or functional structure of pre- and post-synaptic activity into bAP-removal methods may facilitate more accurate inference of input-output correlations. Studies of the functional properties of dendritic spines in visual cortex have excluded spines from analysis where removal of the bAP-component is suspect, using inclusion criteria based on the shape of receptive fields of synaptic inputs, or the magnitude of correlation with output (*Iacaruso et al., 2017*; *Wilson et al., 2016*). We did not apply any such constraints on the task-related responses of inputs to ALM neurons. However, future bAP subtraction methods could use other a priori information about temporal structure of input and output activity as constraints. These constrains could include the binary nature of spikes, the distribution of mean firing rates, and the relative refractory period. These priors could be used by nonlinear inference methods to more accurately disambiguate pre-and post-synaptic activity (J. Yan, A. Kerlin, L. Aitchison, K. Svoboda, S. Turaga, *Cosyne Abstr.* 2018).

## Multi-branch events are coincident with the majority of calcium transients in dendritic branches

The independence of local dendritic spikes varies across models of dendritic computation. In some models, individual dendritic branches can behave as largely independent functional subunits capable of generating local dendritic spikes with a functional selectivity that is distinct from the somatic selectivity (*Archie and Mel, 2000*). Alternatively, facilitative interactions across branches (*Major et al., 2013*; *Major et al., 2008*) or network-level high input states (*Polsky et al., 2009*) could favor the generation of local dendritic spikes that are synchronous across the dendritic tree and reliably accompanied by somatic spiking. The relative independence of local dendritic spikes will depend on many factors, including electrotonic distance between spike-generating sites and the precise spatiotemporal pattern of excitatory and inhibitory input (*Archie and Mel, 2000*; *Behabadi and Mel, 2014*; *Gidon and Segev, 2012*; *Ujfalussy et al., 2018*).

Our recordings from dendrites in motor cortex during a tactile decision-making task are not consistent with models that posit high rates of local dendritic spiking in the absence of global events. At reasonable thresholds for detecting calcium transients, we found almost no independent activity in L2/3 dendrites and a low rate of independent activity in the L5 tuft (*Figure 3C*). It is possible that our estimate of independent activity in the L5 tuft is biased upward by low detectability of calcium transients produced by single sodium APs near the main apical bifurcation (*Helmchen et al., 1999*). However, much of the 'independent' activity in the L5 tuft took the form of branch-specific, sustained activity (up to seconds) following global events (*Figure 2—figure supplement 1*), similar in spatial spread and dynamics to post-plateau potentials that have been measured in ex vivo experiments (*Antic et al., 2010*; *Milojkovic et al., 2007*). In contrast, another study in motor cortex reported that at least 95% of calcium spikes in the L5 tuft were not shared across sibling branches during a forced-running paradigm (*Cichon and Gan, 2015*). This difference may reflect the fact that global events (bAP or tuft calcium spike) may not reliably invade all branches (*Hill et al., 2013*; *Spruston et al., 1995*) or technical differences in the recording and analysis of dendritic events. Alternatively, differences in the learning demands placed upon cortical circuits may affect the prevalence of independent local dendritic spikes. Our results, however, are similar to studies in sensory cortex (*Smith et al., 2013*) and hippocampus (*Sheffield and Dombeck, 2015*), in which the majority of dendritic spikes are coincident with the generation of somatic spikes.

Some branches in the apical tuft of layer five neurons maintained activity for 100's of milliseconds after a global event. What could be the computational roles of these long-lasting events? ALM is a key hub in a multi-regional neural circuit that sustains memory-related activity (*Guo et al., 2014b*). This sustained activity is internally maintained by attractor dynamics in recurrent connections (*Inagaki et al., 2019*), involving ALM, thalamus (*Guo et al., 2017*) and other brain regions (*Gao et al., 2018*). Biophysical models have shown that maintenance of memory-related activity requires time constants of excitation that outlast feedback inhibition (*Wang, 1999*). Long-lasting dendritic excitation could be part of the cellular mechanism of short-term memory.

It is possible that we underestimate the frequency of independent local dendritic spikes if the calcium transients they generate during behavior are well below our measurement noise. In ex vivo (*Antic et al., 2010*; *Golding et al., 2002*; *Milojkovic et al., 2007*; *Nevian et al., 2007*) and in vivo (*Svoboda et al., 1999*) experiments, however, the peak depolarization and calcium influx generated by local dendritic spikes are comparable or larger than the peaks generated by a bAP or calcium spike. Artificial generation of a local dendritic spike on a single branch does not reliably drive somatic action potentials or calcium spikes (*Larkum et al., 2009*; *Losonczy and Magee, 2006*; *Palmer et al., 2014*). Hence, our results suggest that during the tactile decision task, local dendritic spikes are either very rare or are almost always part of a synchronized, multi-branch depolarization of the dendrites that reliably drives somatic action potentials or calcium spikes.

The rarity of independent local dendritic spiking does not preclude functional compartmentalization within the dendrite. Local synaptic input may modulate the depolarization and calcium influx into the dendritic shaft produced by bAPs or calcium spikes (*Larkum et al., 1999b*; *Stuart and Häusser, 2001*; *Waters et al., 2003*), as well as the probability of coincident local dendritic spikes.

## Organization of task-related signals in the dendritic tree

Task-related calcium signals were clustered within the dendritic tree of neurons in motor cortex (*Figures 5* and *7*). Functional similarity among spines and dendrite segments followed an exponential decay with a length constant of approximately 10 µm. Previous work in mouse visual cortex has also observed a distance dependence of retinotopic similarity among spines (*Iacaruso et al., 2017*). In ferret visual cortex, this distance-dependence exhibited an even smaller length constant (5 um; *Scholl et al., 2017*). The short exponential decay of distance-dependent correlations could reflect a clustering of pre-synaptic inputs with similar functional properties or input onto nearby spines from the same axon (*Bloss et al., 2018*; *Kasthuri et al., 2015*; *Knott et al., 2006*). The length constant (~10 µm) is similar to a number of spatially restricted plasticity mechanisms. When activated by stimulation of a single spine, small GTPases spread throughout ~5–10 µm of the dendrite (*Harvey et al., 2008*; *Murakoshi et al., 2011*; *Nishiyama and Yasuda, 2015*) and influence plasticity at nearby spines (*Harvey and Svoboda, 2007*). During development, ryanodine-sensitive calcium release (*Lee et al., 2016*) and BDNF-mediated synaptic depression (*Winnubst et al., 2015*) can produce selective stabilization of inputs with similar spontaneous activity over distances of 5–10 µm. This

length scale is also consistent with the high calcium zone produced by local dendritic spikes (*Major et al., 2008*) and subthreshold nonlinear NMDA receptor-mediated amplification of synaptic calcium signals by the activity of neighboring spines (*Harnett et al., 2012*; *Weber et al., 2016*).

Our analysis allowed us to measure pairwise correlations across branches and spanning large distances within the dendritic tree. We found that in addition to a short exponential decay, correlations exhibited a linear decay over longer distances (up to 180 μm). This was true of both signal and noise correlations. This decay may reflect a combination of multiple processes, including passive and active (via voltage-gated potassium channels) attenuation of both subthreshold potentials and locally initiated dendritic spikes (*Gasparini et al., 2004*; *Harnett et al., 2013*; *Major et al., 2008*; *Milojkovic et al., 2007*).

We also found that the branching structure of dendrites further compartmentalizes task-related signals within the dendrites. A number of mechanisms could confine excitability in branch-specific manner, including current sinks towards branch points (*Branco et al., 2010*; *Marlin and Carter, 2014*; *Müllner et al., 2015*), the sub-branch organization of inhibitory input (*Bloss et al., 2016*), and the distribution of Kv4.2 potassium channels (*Losonczy et al., 2008*).

Diverse behavior-related signals were distributed throughout the dendritic arbor and were compartmentalized by dendritic distance and branching. This compartmentalization may reflect local dendritic operations that expand the processing capacity of individual neurons (*Archie and Mel, 2000*; *Poirazi and Mel, 2001*). Understanding how these operations transform pre-synaptic information may be critical to interpreting the structure and function of cortical circuits. These local operations may also play a critical role in learning (*Frick et al., 2004*; *Guerguiev et al., 2017*; *Losonczy et al., 2008*; *Urbanczik and Senn, 2014*), and future work in the motor cortex could explore the relationship between compartmentalized anatomical changes in the dendrite (*Chen et al., 2015*; *Fu et al., 2012*; *Yang et al., 2009*) and clustered task-related activity during learning.

# Materials and methods

## Key resources table

| Reagent type (species) or resource | Designation | Source or reference | Identifiers | Additional information |
|---|---|---|---|---|
| Genetic reagent (*M. musculus*) | Syt17 NO14 (Mouse) | GENSAT | MMRRC Cat# 034355-UCD, RRID: MMRRC_034355-UCD | |
| Genetic reagent (*M. musculus*) | CamK2a-tTA | JAX | IMSR Cat# JAX:007004, RRID: IMSR_JAX:007004 | |
| Genetic reagent (*M. musculus*) | Ai93 | JAX | IMSR Cat# JAX:024103, RRID: IMSR_JAX:024103 | |
| Genetic reagent (*M. musculus*) | Chrna2 OE25 | GENSAT | MMRRC Cat# 036502-UCD, RRID: MMRRC_036502-UCD | |
| Genetic reagent (*M. musculus*) | ZtTA | JAX | IMSR Cat# JAX: 012266, RRID: IMSR_JAX:012266 | |
| Software and Algorithms | Matlab | Mathworks | RRID:SCR_001622 | |
| Software and Algorithms | ScanImage | Vidrio | RRID:SCR_014307 | |
| Software and Algorithms | Neuromantic | University of Reading | RRID:SCR_013597 | |
| Software and Algorithms | Thunder | Janelia | RRID: SCR_016556 | |
| Software and Algorithms | Spark | Apache | RRID: SCR_016557 | |
| Other | MIMMS microscope 1.0 (2016) | Janelia | RRID:SCR_016511 | |
| Other | Tip-Tilt-Z Sample Positioner | Janelia | RRID:SCR_016528 | |

## Animals

All procedures were in accordance with protocols approved by the Janelia Institutional Animal Care and Use Committee. Triple transgenic mice (both male and female) sparsely expressing GCaMP6f in a subset of layer 2/3 (Syt17 NO14 x CamK2a-tTA x Ai93; MGI:4940641 x JAX:007004 x JAX:024103) and layer 5 (Chrna2 OE25 x ZtTA x Ai93; MGI:5311721 x JAX:012266 x JAX:024103), were housed in a 12 hr:12 hr reverse light:dark cycle. We never observed seizures in these mice, as has been reported for Emx1-Cre x Camk2a-tTa x Ai93 crosses (*Steinmetz et al., 2017*). Surgical procedures were performed under isoflurane anesthesia (3% for induction, 1–1.5% during surgery). A circular (~2.5 mm diameter) craniotomy was made above left ALM (centered at 2.5 mm anterior and 1.5 mm lateral to bregma). A window (triple #1 coverglass 2.5/2.5/3.5 mm diameter; Potomac Photonics, Baltimore, MD) was fixed to the skull using dental adhesive (C and B Metabond; Parkell, Edgewood, NY). A metal bar for head fixation was implanted posterior to the window with a metal loop surrounding the window using dental acrylic. After the surgery, mice recovered for 3–7 days with free access to water. Then, mice were water restricted to 1 mL daily. Training started 3–5 days after the start of water restriction. On days of behavioral training, mice were tested in experimental sessions lasting 1 to 2 hr where they received all their water. Unless otherwise noted in the figure legend, N = 14 neurons, seven mice for figure panels regarding layer 2/3 neurons and N = 5 neurons, four mice for figure panels regarding layer five neurons.

## Tactile decision task

Mice solved an object localization task with their whiskers (modified from *Guo et al., 2014a*; *Guo et al., 2014b*). The stimulus was a metal pin (0.9 mm in diameter) mounted on a galvo motor to reduce vibrations. The pole swung into one of two possible positions (*Figure 2A*). The posterior pole position was approximately 5 mm from the center of the whisker pad. The anterior pole position was 4 mm anterior to the posterior position. A two-spout lickport (4.5 mm between spouts) delivered water reward and recorded the timing of licks.

The sample epoch is defined as the time between the pole movement onset to pole retraction onset (sample epoch, 1.2 s total). The delay epoch lasted for another 2 s after the beginning of pole retraction (delay epoch, 2 s total). An auditory 'response' cue indicated the end of the delay epoch (pure tone, 3.4 kHz, 0.1 s duration). Licking early during the delay period resets the delay-period timer (2 s). Licking the correct lickport after the auditory 'response' cue led to a small drop of water reward. Licking the incorrect lickport was not rewarded nor punished. Trials in which mice did not lick after the 'response' cue were rare and typically occurred only at the end of a session. Animals were trained daily until they reached ~70% correct. Thereafter behavior was combined with imaging (typically 20–40 days after surgery).

## Microscope design

Ultrafast pulses (<100 fs, center wavelength: 960 nm) from a Ti:Sapphire laser (Mai Tai HP; Spectra Physics, Santa Clara, CA) passed through a Pockels Cell (302RM controller with a 350–80 cell; Conoptics, Danbury, CT) to control power. Group delay dispersion (GDD) was pre-compensated by a custom single-prism pulse compressor (*Akturk et al., 2006*; *Kong and Cui, 2013*). Steering and expansion optics directed the beam to an 8 kHz resonant mirror (x-axis, CRS8KHz; Cambridge Technology, Bedford, MA) conjugated to additional x-axis and y-axis galvanometer scanners (5 mm model 6215HSM40, Cambridge Technology). Following the scanning optics, the horizontally-polarized beam entered a remote focus (RF) unit (*Botcherby et al., 2008*; *Botcherby et al., 2012*). Within this unit, the beam passed through a polarizing beam splitter (PBS, PBS251; Thorlabs, Newton, NJ), quarter wave plate (AQWP10M-980, Thorlabs) and tube lens to an objective (CFI Plan Apochromat Lambda 20x Objective Lens NA 0.75 WD 1.00 MM; Nikon, Japan). This objective focused the beam onto a mirror (PF03-03-P01, Thorlabs) mounted on an actuator (LFA1007 voice coil; Equipment Solutions, Sunnyvale, CA). The mirror reflected the beam back through the unit and the polarizing beam splitter redirected the vertically polarized beam towards the imaging objective (25x, 1.05NA, 2 mm working distance; Olympus). A primary dichroic (FF705-Di01−25 × 36; Semrock, Rochester, NY) reflected fluorescence to a second dichroic (565DCXR-cust. Size; 35.5 × 50.2×2.5, r-410–550, t-580–1000 nm, laser grade with ar-coating; Semrock) that separated emission light into green (BG39 and 525/70 nm filters with a H10770PB-40 PMT; Hamamatsu, Japan) and red (not used) channels. The

signal was digitized (NI 5734; National Instruments, Austin, TX) and an image was formed on a FPGA (PXIe-7961R on a PXIe-1073 chassis; National Instruments) controlled by ScanImage 2017 (Vidrio Technologies, Ashburn, VA). Further details of the microscopes core components are available online (RRID: SCR_016511; https://www.flintbox.com/public/offering/4374/). A custom Tip-Tilt-Z Sample Positioner (RRID: SCR_016528; https://www.flintbox.com/public/project/31339/) was used to position the mouse such that the cranial window was perpendicular to the imaging axis.

### Reference volume imaging

Before imaging during behavior, a reference volume for the field-of-view (FOV) was acquired. Dendrite tracing required a reference volume with high SNR and minimal brain motion artefact. To achieve this, we repeatedly imaged the FOV when the mouse was not behaving. We collected 100 image stacks (12 to 20 seconds / volume) at 1x zoom (frames: 525 µm x 525 µm, 1024 pixel x 1024 pixel) from the pia and down to the cell body of interest in 1.6 um steps (~320 um for Syt17 NO14 mice and up to 500 um for Chrna2 OE25 mice). Cross-correlation based registration of stacks to the mean of the most correlated stacks (iteratively: top 30%, then top 70%) removed motion artefacts. All stacks were then averaged to generate the final reference volume.

### Cell selection

Soma locations within the reference volume were manually identified and those coordinates were provided to the imaging software. One to 31 somas or layer five apical trunks were imaged during task performance. A 40 µm x 20 µm imaging frame was centered on each soma or trunk. Registration was done by iterative cross-correlation. Trial-averaged fluorescence was computed for each soma. Cells were selected for functional imaging of the dendrite based on two qualitative criteria: modulation of the signal by the task and sufficient baseline fluorescence to trace dendrites.

### Targeted imaging of dendrites

Tracing of dendrites was done using Neuromantic (*Myatt et al., 2012*) in Semi-Auto mode. Tracing data was loaded to a custom Matlab GUI that enabled selection of different combinations of dendrite branches for targeted imaging. All frames in an imaging sequence were 24 µm x 12 µm (72 pixels x 36 pixels) and had a duration of approximately 2 ms. Average power post-objective varied with imaging depth and ranged from 22 mW to 82 mW. The dendritic length imaged for each session was limited to maintain a sequence rate of approximately 14 Hz (see *Figure 1B,C,D* and *Figure 1F, G,H* for example sessions). Frame positions were calculated to completely cover the volume of the selected dendrite (treating each frame as a 24 µm x 12 µm x 3 µm volume) while minimizing both the total number of frames in the sequence and the predicted acceleration along the z-axis (our slowest axis). Additionally, before each imagining session a closed loop iterative optimization of the z actuator (voice coil) control signal was performed (similar to *Botcherby et al., 2012*). The z trajectory and field placements were then transferred to ScanImage using the MROI API.

### Dendrite image registration and time course extraction

We developed a new process to obtain non-rigid registration of a sequence of small imaging frames irregularly distributed throughout space (see *Boaz, 2019* for code and example session; copy archived at https://github.com/elifesciences-publications/SpineImagingALM). SNR and amount of dendrite in frames varied widely and, thus, independent frame-by-fame registration did not achieve good results. One registration target image per frame was also not suitable, as even small axial motion could be confused with a lateral translation of a thin 3D structures such as dendrites. To address these issues, we developed a multi-step registration and time course extraction procedure in python leveraging open source parallel computing tools (Thunder, RRID: SCR_016556; Apache Spark, RRID: SCR_016557).

Registration included the following steps: initial registration target selection, frame by frame clustering and registration (rigid lateral registration), re-clustering of this laterally aligned data and estimation axial positions to obtain multiple registration targets, and registration of frames to the appropriate axial target (see *Figure 1—figure supplement 2* and *Figure 1—video 1*). The initial target for registration was selected by k-means clustering (30 clusters) on the first 40 PCA components of the complete imaging sequence. Averages of the four largest groups were visually inspected to

select the registration target *Figure 1—figure supplement 2C*). K-means clustering was then used to independently group samples of each frame from the sequence (first 50 components, 200–800 clusters per frame; *Figure 1—figure supplement 2D*) to increase the SNR prior to registration. For each group of samples of each frame, the complete imaging sequence at those samples was averaged and registered to the initial target using cross-correlation (*Figure 1—figure supplement 2E*). Shifts calculated from this registration were then used to constrain (by minimizing brain velocity) the registration of each frame group average to the initial target. Hyper-parameters controlling the balance between cross-correlation peaks and brain velocity constraints in determining shift were optimized (via differential evolution) to maximize the sharpness of the average registered sequence. Lateral shifts calculated for each frame group average were then applied to all individual samples in the group. This laterally-registered data was again clustered (45–90 clusters based on 80 PCA components). These clusters reflected different axial positions as well as activity. A Traveling Salesman Problem solver (https://github.com/dmishin/tsp-solver) ordered these clusters, minimizing the total distance (dissimilarity) between adjacent groups. Adjacent groups were collapsed down to between 4 and 8 (median 5) final groups representing different axial positions (*Figure 1—figure supplement 2F*). The mean of each group served as the registration target for a final registration of all samples, frame-by-frame, belonging to the group. Registration was constrained again by brain velocity parameters, as described above. Thus, for every sample and every frame we determined x and y shifts for lateral motion and z group for axial motion (*Figure 1—figure supplement 2G*).

For high-resolution dendrite and dendritic spine tracing, all frames from a session were projected into 3D space and averaged, taking into account the estimated axial position of each sample and the PSF of the microscope. Dendrite centerline was traced using Neuromantic. This centerline was then dilated in 3D (2.5 μm in z; lateral dilation was based on the estimated radius from the tracing) and divided into 30 μm (*Figures 2* and *3*, *Figure 2—figure supplement 1*, *Figure 4—figure supplement 1*) or approximately 3 μm (*Figures 4*, *5*, *7* and *8*) segments (dendritic masks). Spines were segmented in 3D using a custom-built, semi-automated Matlab GUI (spine masks). The inverse transform of the frames to 3D space was used to extract the fluorescence ($\Delta f$) time course for each mask (dendritic segment or spine). Putative axonal boutons (identified by a 'bead on a string' appearance in the mean volume or during activity) adjacent to the dendrite were segmented but excluded from further analysis.

Baseline was estimated as the mode of a Gaussian kernel density estimator fit to the distribution of F values for a segment in a 2000 sample (~ 2 minute) sliding window. Estimated axial position of each sample was used to correct this baseline estimate for axial motion. Because time courses of individual pixels have Poisson statistics (photon shot noise is the main source of noise), the noise floor (expressed in $\Delta f/f_0$) follows $\frac{\sqrt{N}}{N}$ where $N$ is the number of photons collected in one sample. We calculated $N$ as $m * f$, where $f$ is the total fluorescence collected (arbitrary digital units) and m is the slope of a linear fit to the variance versus the mean of all pixels in the imaging sequence. This noise estimate was also adjusted sample-by-sample for the effects of changes in axial position.

## Independent activity estimation

The signal used to estimate the independent activity was the $\Delta f/f_0$ without bAP subtraction. For layer 2/3 sessions (N = 23) the reference signal was the soma and for layer 5 (N = 16) the apical trunk. Sessions in layer 2/3 where the soma was not imaged were excluded (N = 29). Given a threshold value for a signal we calculated the probability that the signal is above that threshold:

$$P(t) = \frac{1}{2}\left[1 + erf\left(\frac{x(t) - u}{\sigma\sqrt{2}}\right)\right]$$

$$erf(y) = \frac{2}{\sqrt{\pi}}\int_0^y e^{-z^2} dz$$

where $x(t)$ is the $\Delta f/f_0$ at time $t$, $u$ is the threshold, $\sigma$ is the shot-noise estimate.

A false positive probability ($P_{FPR}$) was computed using the same function but $x(t)$ is fixed at the threshold and $u = 0$. We defined the independent activity probability as:

$$P_{independent}(t) = P_{mask}(t) * \left(1 - P_{ref}(t)\right)$$

where $P_{mask}$ is the probability the spine or dendrite signal is above threshold and $P_{ref}$ is the probability the reference signal is above threshold.

We defined the co-activity probability as:

$$P_{co-active}(t) = P_{mask}(t) * P_{ref}(t).$$

False positive rates for independent ($P_{FPR,independent}(t)$) and co-active ($P_{FPR,co-active}(t)$) probabilities were calculated in the same manner, but using the $P_{FPR}$ for mask and reference. Finally, we calculated the proportion independent from averages of probabilities across all timepoints in the recording session:

$$Proportion\ independent = \frac{\overline{P_{independent}}(t)}{\overline{P_{independent}}(t) + \overline{P_{co-active}}(t)}$$

To estimate the false positive (due to shot noise) proportion independent:

$$Proportion\ independent\ FPR = \frac{(\overline{P_{FPR,independent}}(t) - \overline{P_{FPR,co-active}}(t))}{(\overline{P_{co-active}}(t) + \overline{P_{FPR,independent}}(t))}$$

To exclude independent activity preceding or following the reference (*Figure 3—figure supplement 2*), every value in $P_{ref}$ was replaced by the maximum value within a preceding ('after reference') or following ('before reference') time window ('exclusion window'). Proportion independent was then calculated as described above.

## Back-propagating action potential subtraction

We used the soma (N = 23, L2/3 sessions), proximal dendrite (N = 29, L2/3 sessions, 15% most proximal of total dendrite), or apical trunk (N = 16, L5 tuft sessions) as a reference for global activity (putatively dominated by bAPs). The $\Delta f / f_0$ of this reference was then processed with a constrained deconvolution spike inference algorithm (*Pnevmatikakis et al., 2016*; *Vogelstein et al., 2010*; https://github.com/epnev/constrained_foopsi_python) with autoregressive order of 1 and 'fudge factor' of 0.5 (*Figure 4—figure supplement 1A*). The time course for each spine and dendrite mask was fit (by differential evolution minimization of the L2-norm) to the model:

$$model(t) = a \times ref(t) * e^{\frac{-t}{\tau}}$$

$$\underset{a, \tau}{Minimize} \sum_{t=1}^{N} \sqrt[2]{(model(t) - mask(t))^2}$$

where $a$ is an amplitude constant, $ref(t)$ is the deconvolved reference $\Delta f / f_0$, $mask(t)$ is the $\Delta f / f_0$ of the spine or dendrite as function of time and $\tau$ is the time constant of a single-exponential decay kernel (*Figure 4—figure supplement 1B, C*). The residual of this fit was the bAP subtracted signal (*Figure 4—figure supplement 1D*).

Alternative bAP subtraction methods were evaluated in the simulations shown in *Figure 6—figure supplements 1* and *2* (see below for simulation details). Rapid, negative fluorescence transients are inconsistent with the dynamics of GCaMP6, and thus likely represent bAP subtraction errors. 'Alternative 1: Non-negative fit' was the same as the method described above, but instead of minimizing the L2-norm we developed an objective function that penalized negative residuals beyond those expected from photon shot-noise:

$$r(t) = model(t) - mask(t)$$

$$y(t) = \begin{cases} -log(cdf^{neg}(r(t))), & r(t) < 0 \\ -log(cdf^{pos}(r(t))), & r(t) \geq 0 \end{cases}$$

$$\underset{a, \tau}{Minimize} \sum_{t=1}^{N} y(t)$$

where $r(t)$ is the residual at time $t$ of spine or dendrite signal $mask(t)$, $cdf^{neg}$ is a cumulative distribution function estimated from the negative values of the $\Delta f/f_0$ trace for each spine without subtraction and $cdf^{pos}$ is a cumulative distribution function estimated from the positive values of the $\Delta f/f_0$ trace for each spine with subtraction computed by using the 'Alternative 2: Regression – Soma Referenced' subtraction method (see below). Although this approach generated bAP-subtracted traces with 'cleaner' appearance (less negative deflections; data not shown), it still produced inaccurate input-output correlations in our simulations (*Figure 6—figure supplement 1*). For 'Alternative 2: Regression – Soma Referenced' we used the residuals of a robust regression of the spine $\Delta f/f_0$ versus soma $\Delta f/f_0$. For 'Alternative 3: Regression – Dendrite Referenced' we used the residuals of a robust regression of the spine $\Delta f/f_0$ versus the $\Delta f/f_0$ of the closest 30 μm dendrite segment (*Wilson et al., 2016*; *Chen et al., 2013*; *Scholl et al., 2017*). For 'Alternative 4: Least-Squares Fit – Dendrite Referenced' we used the deconvolution subtraction method described above (also in *Figure 4—figure supplement 1*), but the $\Delta f/f_0$ of the closest 30 μm dendrite segment was used as the reference signal instead of the soma.

## Task-associated selectivity

Trials were divided into five epochs (*Figure 4*): sample (pole within reach of whiskers, 1.25 s), early and late delay (1 s increments from end of sample to response cue), and early and late response (1 s increments from response cue). Only correct trials were included. Incorrect trials and trials during which the mouse licked before the go cue were excluded. To test for any task-associated selectivity, we averaged the signal ($\Delta f/f_0$ during each epoch, separating correct left and correct right trials, and performed a nonparametric ANOVA (10 groups total: 5 epochs x two trial-types; p is the proportion of F-statistics from 1000 shuffles of epoch greater than the observed F-statistic). For all significance testing, the number of samples averaged from each epoch was equal (approximately 1 s worth of samples were randomly drawn from the sample epoch).

To visualize and quantify epoch selectivity, we averaged the mean responses during sample, delay (early and late) and response (early and late) epochs across both left and right trials. These mean responses were treated as the magnitudes of three vectors separated by 120° in a polar coordinate system (see *Figures 5B*, *6B* and *8A*). The angle of the vector average provided a one-dimensional representation of the epoch selectivity. Standard error in this angle for each segment was the circular standard deviation of 1000 bootstrap iterations (resampling trials). The significance of differences in epoch angle (Δ angle) between segments was determined by permutation test (1000 shuffles of trial identity, p is the proportion of Δ angles greater than the observed Δ angle). For the circular mean epoch angle across segments, the 95% confidence interval was calculated from 1000 bootstrap iterations (resampling segments).

To visualize and quantify trial-type selectivity, we tested each epoch for significant differences in response during left versus right trials by permutation test (p<0.05, with Bonferroni correction for five null hypotheses). We divided segments into three categories: significantly larger responses during right trials only, left trials only, or both left and right trials depending on epoch.

## Spatial structure of pairwise correlations

Signal correlation between pairs of segments was the Pearson correlation between the mean responses to all 10 conditions (5 epochs x two trial-types) for each segment. To isolate signal correlations from noise correlations, we randomly selected and averaged a non-overlapping 50% of trials for each segment. This was repeated 100 times and the resulting correlation coefficients were averaged. Noise correlation between pairs of segments was the mean noise correlation (Pearson correlation between responses across trials within one condition) across all conditions.

For spines or dendrite segments that were connected by the dendrite imaged within a session, traversal distance was directly calculated from the high-resolution session-based reconstruction. However, for spines and dendrite that connected via dendrite that was not imaged as part of that session's imaging sequence, calculating traversal distance required precise alignment of the imaging session (with segmentation of dendrite and spines) back to the reference space (containing the complete reconstruction of the dendritic tree). A multi-resolution approach from SITK (*Lowekamp et al., 2013*; *Yaniv et al., 2018*) was used to fit a 3d affine transformation from the session space to the FOV space. Center points of spines and dendrite segments were transformed to the reference space

and the closest point along the tracing of dendrite within reference space was determined. Traversal distance (distance of the shortest connecting path through the dendritic tree) could then be calculated from the reference reconstruction.

To quantify mean pairwise correlation as a function of traversal distance, pairs were divided into 11 exponentially spaced distance bins (edges: 0, 2.7, 4.5, 7.4, 12, 20, 33, 55, 90, 148, 245 μm). To calculate the standard error of the mean (SEM), we randomly drew pairs without replacement of the pair members, then the SEM was determined from the standard deviation of pair correlations and the number of pairs drawn for each bin. To average out variation across draws, the reported SEM is the mean of 100 repetitions of this process. Mean within-bin correlation was fit (by differential evolution minimization of the L2-norm) to a 4-parameter model:

$$y = A \times e^{-x/\lambda} + x \times L + B$$

where $x$ is the traversal distance between a pair, $A$ is the amplitude of distance dependent correlation, $\lambda$ is the length constant of exponential decay, $L$ is slope of the linear component, and $B$ is the baseline correlation. Error in $\lambda$ was estimated as the standard deviation of 300 bootstrap iterations (resampling across sessions).

## Simulations

Simulations were designed to analyze the robustness of various measures of dendritic calcium signals to subtraction approaches and variations in simple spike-to-calcium transformations. Simulations were not intended to be biophysically realistic. For each simulation, the geometry of spines and soma, sample rate, and total duration were derived from an actual imaging session. We ran three models of spike-to-calcium transformations with increasing complexity: 'Linear', 'Indicator Nonlinear', and Full Nonlinear. The following describes the Full Nonlinear model as the other models were simplifications thereof. We generated Poisson input (spine) and output (soma) spike trains with a wide distribution pairwise correlations (range of ρ: ∼ 0 – 0.8, mean rate: 1.5 Hz) that were also traversal-distance dependent. This was accomplished by generating a random positive-semidefinite covariance matrix $q$, then adjusting it to be traversal-distance dependent according to:

$$l = q(1 - a^{pre}) + U a^{pre}$$

$$U_{i,j} = exp^{\left(\frac{-x_{i,j}}{\lambda^{pre}}\right)}$$

where $a^{pre}$ is the amplitude of traversal-distance dependent correlations and $U$ is a weight matrix in which $x_{i,j}$ is the traversal distance between spines $i$ and $j$, and $\lambda^{pre}$ is the length constant of pre-synaptic correlations.

We then calculated the positive-semidefinite matrix with unit diagonal that is closest to $l$ (**Higham, 2002**) and used this covariance matrix to specify correlated Poisson spike trains (**Macke et al., 2009**). To implement input lags (**Figure 6—figure supplement 1**), spine spike trains were temporally shifted with respect to the soma spike train. From these spike trains, we calculated a linear component of depolarization at spine $i$:

$$p_i(t) = a^s e_i^s(t) + a^c e^c(t) + a^{coop} \sum_{j \in N(i)} w_{i,j} a^s e_i^s(t)$$

$$w_{i,j} = exp^{\left(\frac{-x_{i,j}}{\lambda^{coop}}\right)}$$

where $a^s$ is the depolarization produced by one pre-synaptic spike, $e_i^s(t)$ are the events (spikes) at spine $i$, $a^c$ is the depolarization produced by the bAP, $e^c(t)$ are the events (spikes) at the cell body (soma), $a^{coop}$ controls the overall cooperativity between spines, and $w_{i,j}$ is the distance-dependent cooperativity between spines $i$ and $j$ given in which $x_{i,j}$ is the traversal distance between spines $i$ and $j$, and $\lambda^{coop}$ is the length constant of cooperativity.

We then calculated a nonlinear component at spine $i$:

$$q_i(t) = m_i(t) \sum_{t'=0}^{\tau^{nl}} h\left(t'\right) e_i^s\left(t - t'\right)$$

$$m_i(t) = a^s \left( \frac{p_i(t)^{n^{nl}}}{p_i(t)^{n^{nl}} + (K^{nl})^{n^{nl}}} \right)$$

$$h\left(t'\right) = exp^{-\left(\frac{-t'}{\tau^{nl}}\right)},$$

where $m_i(t)$ represents nonlinear voltage-dependent unblocking of channels given in which $n^{nl}$ is a cooperativity coefficient and $K^{nl}$ is the depolarization of 50% unblock. $h\left(t'\right)$ represents the dynamics of this component given in which $\tau^{nl}$ is the time constant of decay for the nonlinear component.

Calcium in the spine was modeled according to:

$$[Ca]_i(t) = p_i(t) + rq_i(t)$$

where $r$ is the relative strength of the nonlinear component. Calcium was transformed to fluorescence, $f$, by the indictor by convolution linear exponential filter followed by a stationary nonlinearity:

$$f_i(t) = \frac{g_i(t)^{n^g}}{g_i(t)^{n^g} + (K^g)^{n^g}}$$

$$g_i(t) = [Ca]_i(t) * k_i(t)$$

$$k_i(t) = \exp^{-\left(\frac{t}{\tau_i^s}\right)},$$

where $n^g$ is an indicator cooperativity coefficient, $K^g$ represents half-saturation of the indicator, $k_i(t)$ is the convolution kernel in which $\tau_i^s$ is the time constant of decay for spine $i$.

To simulate the potential for localized differences in spine dynamics (for example, on thick versus thin branches) $\tau_i^s$ was traversal distance dependent, drawn from a random multivariate normal distribution with covariance specified by $exp^{\left(\frac{-x_{i,j}}{\lambda^{decay}}\right)}$. The soma time constant of decay was set by $\tau^c$. $f$ was then linearly scaled such that the 99.5 percentile of $f$ (across all spines and samples) matched the 99.5 percentile of dendritic spine $\Delta f/f_0$ typically observed in high SNR real data ($f^{max}$). Finally, Gaussian noise was added to spine ($\sigma^s$)) and soma ($\sigma^c$) fluorescence.

For *Figure 6—figure supplement 1*, the following parameters were used for 'Full Nonlinear': $a^{pre} = 0.5$; $\lambda^{pre} = 10$ m; $a^s = 1$; $a^c = 3$; $a^{coop} = 0.5$; $\lambda^{coop} = 10$ m; $n^{nl} = 2$; $K^{nl} = 5$; $\tau^{nl} = 100$ ms; $r = 3$; $n^g = 2$; $K^g = 12$; $\lambda^{decay} = 32$ μm; $\tau^s = 0.24$ sec (*mean across spines*); $\tau^c = 0.4$ sec; $f^{max} = 16$ $\Delta f/f_0$. Parameters for "Indicator Nonlinear" were the same as 'Full Nonlinear' except: $a^{coop} = 0$; $r = 0$, which effectively removed cooperativity between spines and all nonlinearities except the indicator nonlinearity. Parameters for 'Linear' were the same as "Indicator Nonlinear", except for omission of the stationary nonlinearity step such that $f_i(t) = [Ca]_i(t) * k_i(t)$. One session (derived from one real L2/3 session with soma imaging) was simulated for each combination of transformation and lag in *Figure 6—figure supplement 1*.

For *Figure 6—figure supplement 2*, all parameters were the same as for simulations for *Figure 6—figure supplement 1*, with the following exceptions. For all 'Presynaptic Clustering' simulations $a^{pre} = 0.5$, $a^{coop} = 0$, and $\lambda^{pre}$ was varied as indicated. For all 'Postsynaptic Cooperativity' simulations $a^{coop} = 0.5$, $a^{pre} = 0$, and $\lambda^{coop}$ was varied as indicated. For all simulations in *Figure 6—figure supplement 2* where $\lambda = 0$ is indicated, we set both $a^{pre} = 0$ and $a^{coop} = 0$. For each combination of distance-dependent process, $\lambda$, and transformation we simulated 23 sessions (derived from the real L2/3 sessions with soma imaging).

The cartoon tuning curves, input-output correlations, and the pairwise correlations between spines in *Figure 6* are shown for didactic purposes. They are simplifications of our conclusions from simulation results in *Figure 6—figure supplements 1* and *2*.

## Acknowledgements

We would like to thank Na Ji and Rongwen Lu for advice and assistance on microscopy; Charles Gerfen for the Syt17-NO14-Cre and Chrna2-OE25-Cre mice; Mark Johnson, Tara Srirangarajan, and Rinat Mohar for mouse training and surgery; Jeremy Freeman and Jason Wittenbach for help with parallel computing; Jinyao Yan and Srini Turaga for useful discussions; Jeffrey Magee, Kaspar Podgorski, and Hod Dana for comments on the manuscript. This work was funded by Howard Hughes Medical Institute.

## Additional information

### Competing interests

Karel Svoboda: Reviewing editor, *eLife*. The other authors declare that no competing interests exist.

### Funding

| Funder | Author |
| --- | --- |
| Howard Hughes Medical Institute | Aaron Kerlin<br>Mohar Boaz<br>Daniel Flickinger<br>Bryan J MacLennan<br>Matthew B Dean<br>Nelson Spruston<br>Karel Svoboda |

The funders had no role in study design, data collection and interpretation, or the decision to submit the work for publication.

### Author contributions

Aaron Kerlin, Conceptualization, Data curation, Software, Formal analysis, Validation, Investigation, Visualization, Methodology, Writing—original draft, Writing—review and editing; Boaz Mohar, Data curation, Software, Formal analysis, Validation, Investigation, Visualization, Methodology, Writing—original draft, Writing—review and editing; Daniel Flickinger, Methodology; Bryan J MacLennan, Courtney Davis, Investigation, Methodology; Matthew B Dean, Visualization; Nelson Spruston, Resources, Formal analysis, Supervision, Writing—original draft, Project administration, Writing—review and editing; Karel Svoboda, Conceptualization, Resources, Formal analysis, Supervision, Writing—original draft, Project administration, Writing—review and editing

### Author ORCIDs

Aaron Kerlin https://orcid.org/0000-0002-8257-5216
Boaz Mohar https://orcid.org/0000-0002-8613-2869
Daniel Flickinger http://orcid.org/0000-0002-8361-3122
Nelson Spruston http://orcid.org/0000-0003-3118-1636
Karel Svoboda https://orcid.org/0000-0002-6670-7362

### Ethics

Animal experimentation: All procedures were in accordance with protocols approved by the Janelia Research Campus Institutional Animal Care and Use Committee. IACUC 14-115.

### Decision letter and Author response

Decision letter https://doi.org/10.7554/eLife.46966.sa1
Author response https://doi.org/10.7554/eLife.46966.sa2

## Additional files

### Supplementary files

• Transparent reporting form

### Data availability

All source data can be browsed by the public at https://spinevis.janelia.org/. Registration and analysis code can be found at https://github.com/boazmohar/SpineImagingALM (copy archived at https://github.com/elifesciences-publications/SpineImagingALM).

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
