## [Decision Letter]

Thank you for submitting your article "Functional clustering of dendritic activity during decision-making" for consideration by *eLife*. Your article has been reviewed by two peer reviewers, including a guest Reviewing Editor, and the evaluation has been overseen by Joshua Gold as the Senior Editor. The reviewers have opted to remain anonymous.

The reviewers have discussed the reviews with one another and the Reviewing Editor has drafted this decision to help you prepare a revised submission.

Summary:

This manuscript by Kerlin and colleagues reports advanced approaches for in vivo two-photon calcium imaging of dendritic signals. The authors imaged calcium signals in pyramidal cell dendrites in the anterior lateral motor (ALM) cortex of mice performing a tactile decision task. They developed novel calcium imaging and analysis methods to estimate the spatial structure of activity in spines and dendrites while mice performed this task. They report that isolated dendritic events are rare in these cells and that most dendritic events are 'global', in that calcium transients were detected simultaneously throughout the soma and all of the imaged parts of the dendritic arbor. They also report some spatial clustering of functionally related spines with a length constant of ~10 μm and an additional decay over longer distances. Finally, they also report that the branching structure of dendrites further compartmentalizes task-related signals within the dendrites.

The novel imaging and registration methods are described in great details and provide highly valuable data sets of somato-dendritic calcium dynamics in awake behaving mice. The authors provide detailed analyses and controls to reliably quantify the changes in GCamP6 fluorescence in dendrites and spines. These include both a more efficient sampling of longer stretches of multiple dendrites and a more principled way for bAP subtraction compared to some previous publications. This resource is very valuable for this research field and sends a critical message of caution regarding the analysis and interpretation of dendritic calcium imaging.

While the imaging method and analysis tools are a clear strength of this study, the presentation and interpretation of the data raised some concerns. Despite their message of caution about calcium imaging data analysis, the authors seem to emphasize results that are then described as non-significant (e.g. isolated dendritic activity in L2/3) or for which not enough information is provided to fully assess the nature of these signals (e.g. dendritic functional clustering). It is unclear whether it is a method paper with examples taken from ALM neurons or whether it is a study about ALM neurons function and information processing as the title of the article would suggest. We have the following suggestions to improve the manuscript.

Essential revisions:

1) Isolated dendritic events: the authors should quantify the proportion of truly independent dendritic events that started before or wholly in the absence of global events and the proportion of calcium signals that are a sustained elevation in fluorescence occurring after a global event.

The only truly independent event is shown in Figure 2—figure supplement 1D. Since in the example trace, there is no activity at all in the other compartments it is unclear whether the lack of activity in the other compartments is due to an imaging issue (e.g. movement artefacts) or truly represents a lack of activity for this specific trial.

For the sustained elevation, the authors should clarify whether these cases correspond to transients with a decay time systematically longer for a given compartment, and as such would be detected as 'isolated events' each time the amplitude of the transient was higher than in other compartments. If so, the difference would be a higher response amplitude but not a sustained activity per se. (e.g. plot decay time constant as a function of transients peak amplitude)

2) Task-related Calcium Signals in dendrites. As mentioned in the Introduction, imaging short stretches of dendrite (e.g. Cichon and Gan, 2015; Sheffield and Dombeck, 2014), makes it difficult to disambiguate global, branch-specific, or spine-specific activity. It is thus unclear why the authors decide to analyse 3μm short dendritic segment for assessing the selectivity of responses for behavioral epoch. Such short segments could correspond to a spine that would overlap with the dendritic segment in the imaged focal plane. Why is it that short dendritic segments were included in some cases but apparently not during the characterization of independent activity in Figure 2? How correlated is each small bit of dendritic segment to the overall dendrite? Do larger dendritic segments show any type of selectivity?

For this point as for the rest of the article, authors should provide examples of traces of task-related dendritic signals. Authors should provide some basic characterizations of spine/dendritic activity levels at various stages of training or during different trial types: What is the behavior of trial-type-selective spines during choice error trials? What about epoch-selective spines during trials in which the mouse licked too early?

3) Figure 5: Since the selectivity for trial epoch and trial type is the main result the authors are emphasizing it is crucial to provide traces of these branch specific events as shown for the spines in Figure 4. The color code used in panel B, D, G, I is based on the magnitudes of three vectors: it is hard to relate this 1 dimensional representation of the epoch selectivity to actual fluorescence changes in these dendritic compartments. In this figure, the quantification of the results is only shown for spines. In order to support the claims of dendritic functional selectivity (which is the title of the figure and of the article), the authors have to show the same quantification for dendritic segments (5K to 5P).

Figure 5M and 5P: since authors have compared pairs of spines belonging to each neuron, they should report the mean +/- SEM per neuron.

- Indicate n numbers. Figure 3C and 3D: what is the total number of dendrites/neurons/animals included in this analysis? Same for all figures e.g. Figure 5M, 5P: number of spines/dendrites/neurons/animals; numbers are very different before and after subtraction. For selective spines: were there clustered on a few dendrites, or the number of selective spines per dendrite was normally distributed?

Figure 7: give n numbers and number of data points (binning) for each panel (7C and 7F).

Figure 8: give n, number of pairs<10 um.

4) For bAP subtraction, why not use signal from nearby dendritic segments for more locally accurate estimates of global calcium activity instead of using deconvolved somatic signal? Their analysis of inferred length constants for different subtraction methods in Figure 6—figure supplement 2 is useful, but it does not address the possible use of similarly de- and then re-convolving dendritic signal to get a least squares fit (instead of the naïve linear regression subtraction). Do nearby spines tend to have similarly timed or similar levels of independent activity?

5) The authors are attempting to make this dataset a shared resource for the field by placing the raw data online and sharing all relevant code. While this is undoubtedly a positive initiative that should certainly be supported, the authors have not yet provided the relevant analysis code in the github repository, and the browser (spinevis.janelia.org) was not accessible when the reviewers tried to access it.

---

## [Author Response]

Essential revisions:1) Isolated dendritic events: the authors should quantify the proportion of truly independent dendritic events that started before or wholly in the absence of global events and the proportion of calcium signals that are a sustained elevation in fluorescence occurring after a global event.

In our analysis we define “independent activity” as signal that is not coincident with the reference signal. Our data cannot establish true independence in the causal sense. However, we can extend the window over which activity is considered coincident, as the reviewers suggest. We now show how the proportion independent activity declines as the window of coincidence is extended before or after the reference signal (Figure 3—figure supplement 2.

The only truly independent event is shown in Figure 2—figure supplement 1D. Since in the example trace, there is no activity at all in the other compartments it is unclear whether the lack of activity in the other compartments is due to an imaging issue (e.g. movement artefacts) or truly represents a lack of activity for this specific trial.

The lack of activity in the other compartments is not caused by movement artefacts for the following reasons: a) the other compartments are visible; b) the apical trunk signal is insensitive to axial (difficult to correct) motion because the trunk travels axially. An additional example of independent activity that precedes trunk activity by roughly one second is now shown in Figure 4—figure supplement 2.

For the sustained elevation, the authors should clarify whether these cases correspond to transients with a decay time systematically longer for a given compartment, and as such would be detected as 'isolated events' each time the amplitude of the transient was higher than in other compartments. If so, the difference would be a higher response amplitude but not a sustained activity per se. (e.g. plot decay time constant as a function of transients peak amplitude)

We performed deconvolution of the reference signal with sparse reconstruction and then used this to estimate the amplitude and “systematic” decay time constant for each compartment. As shown in Author response image 1, correlation between amplitude and decay, as well as amplitude and proportion independent are weakly negative, contrary to the hypothesis that isolated events arise mostly from compartments with systematically higher amplitude. There is a weak, but positive correlation between proportion independent and the decay time constant. We do not consider this analysis of sufficient general interest to include in the manuscript.

**Author response image 1. respfig1:** Correlation between decay, amplitude, and proportion independent. Each point is a 30 μm segment of a L5 tuft dendrite.

2) Task-related Calcium Signals in dendrites. As mentioned in the Introduction, imaging short stretches of dendrite (e.g. Cichon and Gan, 2015; Sheffield and Dombeck, 2014), makes it difficult to disambiguate global, branch-specific, or spine-specific activity. It is thus unclear why the authors decide to analyse 3μm short dendritic segment for assessing the selectivity of responses for behavioral epoch. Such short segments could correspond to a spine that would overlap with the dendritic segment in the imaged focal plane. Why is it that short dendritic segments were included in some cases but apparently not during the characterization of independent activity in Figure 2? How correlated is each small bit of dendritic segment to the overall dendrite? Do larger dendritic segments show any type of selectivity?

Small segments of dendrite (3 um) were used so that we could measure the length scale of correlation between dendritic segments over spatial scales of 10 um. In Figure 6 we show that correlations between spines have an exponential decay of around 10 um.

We now provide quantification of dendritic functional selectivity similar to Figure 5K-P, but for large (30 um) segments of dendrite in Figure 5—figure supplement 1G-L.

For this point as for the rest of the article, authors should provide examples of traces of task-related dendritic signals. Authors should provide some basic characterizations of spine/dendritic activity levels at various stages of training or during different trial types: What is the behavior of trial-type-selective spines during choice error trials? What about epoch-selective spines during trials in which the mouse licked too early?

We now provide examples of traces of task-related dendritic signals during both correct and error trials (Figure 4—figure supplement 2).

Mice were not imaged until they reached a high level of performance, thus there were few incorrect trials in most behavioral sessions and uncertainty in response magnitudes were too large to draw conclusions about selectivity. Early licking trials were not stereotyped and, thus, trial-average responses were difficult to interpret.

3) Figure 5: Since the selectivity for trial epoch and trial type is the main result the authors are emphasizing it is crucial to provide traces of these branch specific events as shown for the spines in Figure 4. The color code used in panel B, D, G, I is based on the magnitudes of three vectors: it is hard to relate this 1 dimensional representation of the epoch selectivity to actual fluorescence changes in these dendritic compartments. In this figure, the quantification of the results is only shown for spines. In order to support the claims of dendritic functional selectivity (which is the title of the figure and of the article), the authors have to show the same quantification for dendritic segments (5K to 5P).

We now provide examples of traces of task-related, branch-specific events (Figure 4—figure supplement 2). We quantify dendritic functional selectivity similar to Figure 5K-P, but for 3 μm segments of dendrite (Figure 5—figure supplement 1A-F).

Figure 5M and 5P: since authors have compared pairs of spines belonging to each neuron, they should report the mean +/- SEM per neuron.

Figure 5M and 5P show proportions, not means. “Different” versus “not different” is a binary measure.

- Indicate n numbers. Figure 3C and 3D: what is the total number of dendrites/neurons/animals included in this analysis?

We have added the following to the Materials and methods: “Unless otherwise noted in the figure legend, N = 14 neurons, 7 mice for figure panels on layer 2/3 neurons and N = 5 neurons, 4 mice for figure panels on layer 5 neurons” A note has been added to Figure 3, since it deviates from the above numbers of neurons and mice.

We are not sure what the reviewers mean by number of dendrites. Number of primary dendrites? Branches? We are not sure this is necessary for the figure.

Same for all figures e.g. Figure 5M, 5P: number of spines/dendrites/neurons/animals; numbers are very different before and after subtraction. For selective spines: were there clustered on a few dendrites, or the number of selective spines per dendrite was normally distributed?

We have added these numbers to the figure legend. The distribution of significantly selective spines (irrespective of preferred epoch or trial-type) may be impacted by signal-to-noise ratio and signal-to-background ratio that vary along the dendrite in ways that are difficult to correct. We would be reluctant to draw conclusions from any such analysis.

Figure 7: give n numbers and number of data points (binning) for each panel (7C and 7F).

These numbers have been added to the legend. Bin edges are defined in the Materials and methods as now noted in the figure legend.

Figure 8: give n, number of pairs<10 um.

These numbers have been added in the figure legend.

4) For bAP subtraction, why not use signal from nearby dendritic segments for more locally accurate estimates of global calcium activity instead of using deconvolved somatic signal? Their analysis of inferred length constants for different subtraction methods in Figure 6—figure supplement 2 is useful, but it does not address the possible use of similarly de- and then re-convolving dendritic signal to get a least squares fit (instead of the naïve linear regression subtraction). Do nearby spines tend to have similarly timed or similar levels of independent activity?

We have added de- and re-convolving nearby dendrite signal to get a least squares fit to our simulations of subtraction methods (Figure 6—figure supplement 2C, see “Least-Squares Fit – Dendrite Referenced”). The results are similar to dendrite-referenced linear regression.

5) The authors are attempting to make this dataset a shared resource for the field by placing the raw data online and sharing all relevant code. While this is undoubtedly a positive initiative that should certainly be supported, the authors have not yet provided the relevant analysis code in the github repository, and the browser (spinevis.janelia.org) was not accessible when the reviewers tried to access it.

We apologize for this oversight. The browser is accessible again. We have placed relevant image-processing code in the github repository. All other relevant analysis code will be added by the time the paper is in published.